# GRAPH COARSENING WITH NEURAL NETWORKS

**Chen Cai** [*]        **Dingkang Wang** [†]        **Yusu Wang** [‡]

## ABSTRACT

As large-scale graphs become increasingly more prevalent, it poses significant computational challenges to process, extract and analyze large graph data. Graph coarsening is one popular technique to reduce the size of a graph while maintaining essential properties. Despite rich graph coarsening literature, there is only limited exploration of data-driven methods in the field. In this work, we leverage the recent progress of deep learning on graphs for graph coarsening. We first propose a framework for measuring the quality of coarsening algorithm and show that depending on the goal, we need to carefully choose the Laplace operator on the coarse graph and associated projection/lift operators. Motivated by the observation that the current choice of edge weight for the coarse graph may be suboptimal, we parametrize the weight assignment map with graph neural networks and train it to improve the coarsening quality in an unsupervised way. Through extensive experiments on both synthetic and real networks, we demonstrate that our method significantly improves common graph coarsening methods under various metrics, reduction ratios, graph sizes, and graph types. It generalizes to graphs of larger size ($25\times$ of training graphs), is adaptive to different losses (differentiable and non-differentiable), and scales to much larger graphs than previous work.

## 1 INTRODUCTION

Many complex structures can be modeled by graphs, such as social networks, molecular graphs, biological protein-protein interaction networks, knowledge graphs, and recommender systems. As large scale-graphs become increasingly ubiquitous in various applications, they pose significant computational challenges to process, extract and analyze information. It is therefore natural to look for ways to simplify the graph while preserving the properties of interest.

There are two major ways to simplify graphs. First, one may reduce the number of edges, known as graph edge sparsification. It is known that pairwise distance (spanner), graph cut (cut sparsifier), eigenvalues (spectral sparsifier) can be approximately maintained via removing edges. A key result (Spielman & Teng, 2004) in the spectral sparsification is that any dense graph of size $N$ can be sparsified to $O(N log^c N/\epsilon^2)$ edges in nearly linear time using a simple randomized algorithm based on the effective resistance.

Alternatively, one could also reduce the number of nodes to a subset of the original node set. The first challenge here is how to choose the topology (edge set) of the smaller graph spanned by the sparsified node set. On the extreme, one can take the complete graph spanned by the sampled nodes. However, its dense structure prohibits easy interpretation and poses computational overhead for setting the $\Theta(n^2)$ weights of edges. This paper focuses on *graph coarsening*, which reduces the number of nodes by contracting disjoint sets of connected vertices. The original idea dates back to the algebraic multigrid literature (Ruge & Stüben, 1987) and has found various applications in graph partitioning (Hendrickson & Leland, 1995; Karypis & Kumar, 1998; Kushnir et al., 2006), visualization (Harel & Koren, 2000; Hu, 2005; Walshaw, 2000) and machine learning (Lafon & Lee, 2006; Gavish et al., 2010; Shuman et al., 2015).

However, most existing graph coarsening algorithms come with two restrictions. First, they are *prespecified* and not adapted to specific data nor different goals. Second, most coarsening algorithms set the edge weights of the coarse graph equal to the sum of weights of crossing edges in the original

---

[*]University of California, San Diego, `c1cai@ucsd.edu`

[†]Ohio State University, `wang.6150@osu.edu`

[‡]University of California, San Diego, `yusuwang@ucsd.edu`

graph. This means the weights of the coarse graph is determined by the coarsening algorithm (of the vertex set), leaving no room for adjustment.

With the two observations above, we aim to develop a data-driven approach to better assigning weights for the coarse graph depending on specific goals at hand. We will leverage the recent progress of deep learning on graphs to develop a framework to learn to assign edge weights *in an unsupervised manner* from a collection of input (small) graphs. This learned weight-assignment map can then be applied to new graphs (of potentially much larger sizes). In particular, our contributions are threefold.

- First, depending on the quantity of interest $\mathcal{F}$ (such as the quadratic form w.r.t. Laplace operator), one has to carefully choose projection/lift operator to relate quantities defined on graphs of different sizes. We formulate this as the invariance of $\mathcal{F}$ under lift map, and provide three cases of projection/lift map as well as the corresponding operators on the coarse graph. Interestingly, those operators all can be seen as the special cases of doubly-weighted Laplace operators on coarse graphs (Horak & Jost, 2013).
- Second, we are the first to propose and develop a framework to learn the edge weights of the coarse graphs via graph neural networks (GNN) in an unsupervised manner. We show convincing results both theoretically and empirically that changing the weights is crucial to improve the quality of coarse graphs.
- Third, through extensive experiments on both synthetic graphs and real networks, we demonstrate that our method GOREN significantly improves common graph coarsening methods under different evaluation metrics, reduction ratios, graph sizes, and graph types. It generalizes to graphs of larger size (than the training graphs), adapts to different losses (so as to preserve different properties of original graphs), and scales to much larger graphs than what previous work can handle. Even for losses that are not differentiable w.r.t the weights of the coarse graph, we show training networks with a differentiable auxiliary loss still improves the result.

## 2 RELATED WORK

**Graph sparsification**. Graph sparsification is firstly proposed to solve linear systems involving combinatorial graph Laplacian efficiently. Spielman & Teng (2011); Spielman & Srivastava (2011) showed that for any undirected graph $G$ of $N$ vertices, a spectral sparsifier of $G$ with only $O(Nlog^c N/\epsilon^2)$ edges can be constructed in nearly-linear time. [1] Later on, the time complexity and the dependency on the number of the edges are reduced by various researchers (Batson et al., 2012; Allen-Zhu et al., 2015; Lee & Sun, 2018; 2017).

**Graph coarsening**. Previous work on graph coarsening focuses on preserving different properties, usually related to the spectrum of the original graph and coarse graph. Loukas & Vandergheynst (2018); Loukas (2019) focus on the restricted spectral approximation, a modification of the spectral similarity measure used for graph sparsification. Hermsdorff & Gunderson (2019) develop a probabilistic framework to preserve inverse Laplacian.

**Deep learning on graphs**. As an effort of generalizing convolution neural network to the graphs and manifolds, graph neural networks is proposed to analyze graph-structured data. They have achieved state-of-the-art performance in node classification (Kipf & Welling, 2016), knowledge graph completion (Schlichtkrull et al., 2018), link prediction (Dettmers et al., 2018; Gurukar et al., 2019), combinatorial optimization (Li et al., 2018b; Khalil et al., 2017), property prediction (Duvenaud et al., 2015; Xie & Grossman, 2018) and physics simulation (Sanchez-Gonzalez et al., 2020).

**Deep generative model for graphs**. To generative realistic graphs such as molecules and parse trees, various approaches have been taken to model complex distributions over structures and attributes, such as variational autoencoder (Simonovsky & Komodakis, 2018; Ma et al., 2018), generative adversarial networks (GAN) (De Cao & Kipf, 2018; Zhou et al., 2019), deep autoregressive model (Liao et al., 2019; You et al., 2018b; Li et al., 2018a), and reinforcement learning type approach (You et al., 2018a). Zhou et al. (2019) proposes a GAN-based framework to preserve the hierarchical community structure via algebraic multigrid method during the generation process. However, different from our approach, the coarse graphs in Zhou et al. (2019) are not learned.

---

[1]The algorithm runs in $O(M.polylogN)$ time, where $M$ and $N$ are the numbers of edges and vertices.

# 3 PROPOSED APPROACH: LEARNING EDGE WEIGHT WITH GNN

## 3.1 HIGH-LEVEL OVERVIEW

Our input is a non-attributed (weighted or unweighted) graph $G = (V, E)$. Our goal is to construct an appropriate "coarser" graph $\widehat{G} = (\widehat{V}, \widehat{E})$ that preserves certain properties of $G$. Here, by a "coarser" graph, we assume that $|\widehat{V}| << |V|$ and there is a surjective map $\pi : V \to \widehat{V}$ that we call the *vertex map*. Intuitively, (see figure on the right), for any node $\hat{v} \in \widehat{V}$, all nodes $\pi^{-1}(\hat{v}) \subset V$ are mapped to this *super-node* $\hat{v}$ in the coarser graph $\widehat{G}$. We will later propose a GNN based framework that can be trained using a collection of existing graphs *in an unsupervised manner*, so as to construct such a coarse graph $\widehat{G}$ for a future input graph $G$ (presumably coming from the same family as training graphs) that can preserve properties of $G$ effectively.

We will in particular focus on preserving properties of the *Laplace operator* $\mathcal{O}_G$ of $G$, which is by far the most common operator associated to graphs, and forms the foundation for spectral methods. Specifically, given $G = (V = \{v_1, \ldots, v_N\}, E)$ with $w : E \to \mathbb{R}$ being the weight function for $G$ (all edges have weight 1 if $G$ is unweighted), let $W$ the corresponding $N \times N$ edge-weight matrix where $W[i][j] = w(v_i, v_j)$ if edge $(v_i, v_j) \in E$ and 0 otherwise. Set $D$ to be the $N \times N$ diagonal matrix with $D[i][i]$ equal to the sum of weights of all edges incident to $v_i$. The standard *(unnormalized) combinatorial Laplace operator* of $G$ is then defined as $L = D - W$. The *normalized Laplacian* is defined as $\mathcal{L} = D^{-1/2} L D^{-1/2} = I - D^{-1/2} W D^{-1/2}$.

However, to make this problem as well as our proposed approach concrete, various components need to be built appropriately. We provide an overview here, and they will be detailed in the remainder of this section.

- Assuming that the set of super-nodes $\widehat{V}$ as well as the map $\pi : V \to \widehat{V}$ are given, one still need to decide how to set up the connectivity (i.e, edge set $\widehat{E}$) for the coarse graph $\widehat{G} = (\widehat{V}, \widehat{E})$. We introduce a natural choice in Section 3.2, and provide some justification for this choice.

- As the graph $G$ and the coarse graph $\widehat{G}$ have the different number of nodes, their Laplace operators $\mathcal{O}_G$ and $\mathcal{O}_{\widehat{G}}$ of two graphs are not directly comparable. Instead, we will compare $\mathcal{F}(\mathcal{O}_G, f)$ and $\mathcal{F}(\mathcal{O}_{\widehat{G}}, \hat{f})$, where $\mathcal{F}$ is a functional intrinsic to the graph at hand (invariant to the permutation of vertices), such as the quadratic form or Rayleigh quotient. However, it turns out that depending on the choice of $\mathcal{F}$, we need to choose the precise form of the Laplacian $\mathcal{O}_{\widehat{G}}$, as well as the (so-called lifting and projection) maps relating these two objects, carefully, so as they are comparable. We describe these in detail in Section 3.3.

- In Section 3.4 we show that adjusting the weights of the coarse graph $\widehat{G}$ can significantly improve the quality of $\widehat{G}$. This motivates a learning approach to learn a strategy (a map) to assign these weights from a collection of given graphs. We then propose a GNN-based framework to do so in an unsupervised manner. Extensive experimental studies will be presented in Section 4.

## 3.2 CONSTRUCTION OF COARSE GRAPH

Assume that we are already given the set of super-nodes $\widehat{V} = \{\hat{v}_1, \ldots, \hat{v}_n\}$ for the coarse graph $\widehat{G}$ together with the vertex map $\pi : V \to \widehat{V}$ – There has been much prior work on computing the sparsified set $\widehat{V} \subset V$ and $\pi$ (Loukas & Vandergheynst, 2018; Loukas, 2019); and if the vertex map $\pi$ is not given, then we can simply define it by setting $\pi(v)$ for each $v \in V$ to be the nearest neighbor of $v$ in $\widehat{V}$ in terms of graph shortest path distance in $G$ (Dey et al., 2013).

To construct edges for the coarse graph $\widehat{G} = (\widehat{V}, \widehat{E})$ together with the edge weight function $\hat{w} : \widehat{E} \to \mathbb{R}$, instead of using a complete weighted graph over $\widehat{V}$, which is too dense and expensive, we set $\widehat{E}$ to be those edges "induced" from $G$ when collapsing each *cluster* $\pi^{-1}(\hat{v})$ to its corresponding super-node $\hat{v} \in \widehat{V}$: Specifically, $(\hat{v}, \hat{v}') \in \widehat{E}$ if and only if there is an edge $(v, v') \in E$ such that $\pi(v) = \hat{v}$ and $\pi(v') = \hat{v}'$. The weight of this edge is $\hat{w}(\hat{v}, \hat{v}') := \sum_{(v,v') \in E\left(\pi^{-1}(\hat{v}), \pi^{-1}(\hat{v}')\right)} w(v, v')$ where $E(A, B) \subseteq E$ stands for the set of edges crossing sets $A, B \subseteq V$; i.e., $\hat{w}(\hat{v}, \hat{v}')$ is the total weights of all crossing edges in $G$ between clusters $\pi^{-1}(\hat{v})$ and $\pi^{-1}(\hat{v}')$ in $V$. We refer to $\widehat{G}$ constructed

this way the $\widehat{V}$-*induced coarse graph*. As shown in Dey et al. (2013), if the original graph $G$ is the 1-skeleton of a hidden space $X$, then this induced graph captures the topological of $X$ at a coarser level if $\widehat{V}$ is a so-called $\delta$-net of the original vertex set $V$ w.r.t. the graph shortest path metric.

Let $\widehat{W}$ be the edge weight matrix, and $\widehat{D}$ be the diagonal matrix encoding the sum of edge weights incident to each vertex as before. Then the standard combinatorial Laplace operator w.r.t. $\widehat{G}$ is simply $\widehat{L} = \widehat{D} - \widehat{W}$.

**Relation to the operator of (Loukas, 2019).** Interestingly, this construction of the coarse graph $\widehat{G}$ coincides with the coarse Laplace operator for a sparsified vertex set $\widehat{V}$ constructed by Loukas (2019). We will use this view of the Laplace operator later; hence we briefly introduce the construction of Loukas (2019) (adapted to our setting): Given the vertex map $\pi : V \to \widehat{V}$, we set a $n \times N$ matrix $P$ by $P[r, i] = \begin{cases} \frac{1}{|\pi^{-1}(\hat{v}_r)|} & \text{if } v_i \in \pi^{-1}(\hat{v}_r) \\ 0 & \text{otherwise} \end{cases}$. In what follows, we denote $\gamma_r := |\pi^{-1}(\hat{v}_r)|$ for any $r \in [1, n]$, which is the size of the cluster of $\hat{v}_r$ in $V$. $P$ can be considered as the weighted projection matrix of the vertex set from $V$ to $\widehat{V}$. Let $P^+$ denote the Moore-Penrose pseudoinverse of $P$, which can be intuitively viewed as a way to lift a function on $\widehat{V}$ (a vector in $\mathbb{R}^n$) to a function over $V$ (a vector in $\mathbb{R}^N$). As shown in Loukas (2019), $P^+$ is the $N \times n$ matrix where $P^+[i, r] = 1$ if and only if $\pi(v_i) = \hat{v}_r$. See Appendix A.2 for a toy example. Finally, Loukas (2019) defines an operator for the coarsened vertex set $\widehat{V}$ to be $\tilde{L}_{\widehat{V}} = (P^+)^T L P^+$. Intuitively, $\widehat{L}$ operators on $n$-vectors. For any $n$-vector $\hat{f} \in \mathbb{R}^n$, $\tilde{L}_{\widehat{V}} \hat{f}$ first lifts $\hat{f}$ to a $N$-vector $f = P^+ \hat{f}$, and then perform $L$ on $f$, and then project it down to $n$-dimensional via $(P^+)^T$.

**Proposition 3.1.** *(Loukas, 2019) The combinatorial graph Laplace operator $\widehat{L} = \widehat{D} - \widehat{W}$ for the $\widehat{V}$-induced coarse graph $\widehat{G}$ constructed above equals to the operator $\tilde{L}_{\widehat{V}} = (P^+)^T L P^+$.*

### 3.3 LAPLACE OPERATOR FOR THE COARSE GRAPH

We now have an input graph $G = (V, E)$ and a coarse graph $\widehat{G}$ induced from the sparsified node set $\widehat{V}$, and we wish to compare their corresponding Laplace operators. However, as $\mathcal{O}_G$ operates on $\mathbb{R}^N$ (i.e, functions on the vertex set $V$ of $G$) and $\mathcal{O}_{\widehat{G}}$ operates on $\mathbb{R}^n$, we will compare them by their effects on "corresponding" objects. Loukas & Vandergheynst (2018); Loukas (2019) proposed to use the quadratic form to measure the similarity between the two linear operators. In particular, given a linear operator $A$ on $\mathbb{R}^N$ and any $x \in \mathbb{R}^N$, $\mathsf{Q}_A(x) = x^T A x$. The quadratic form has also been used for measuring spectral approximation under edge sparsification. The proof of the following result is in Appendix A.2.

**Proposition 3.2.** *For any vector $\hat{x} \in \mathbb{R}^n$, we have that $\mathsf{Q}_{\widehat{L}}(\hat{x}) = \mathsf{Q}_L(P^+ \hat{x})$, where $\widehat{L}$ is the combinatorial Laplace operator for the $\widehat{V}$-induced coarse graph $\widehat{G}$ constructed above. That is, set $x := P^+ \hat{x}$ as the lift of $\hat{x}$ in $\mathbb{R}^N$, then $\hat{x}^T \widehat{L} \hat{x} = x^T L x$.*

Intuitively, this suggests that if later, we measure the similarity between $L$ and some Laplace operator for the coarse graph $\widehat{G}$ based on a loss from quadratic form difference, then we should choose the Laplace operator $\mathcal{O}_{\widehat{G}}$ to be $\widehat{L}$ and compare $\mathsf{Q}_{\widehat{L}}(Px)$ with $\mathsf{Q}_L(x)$. We further formalize this by considering the *lifting map* $\mathcal{U} : \mathbb{R}^n \to \mathbb{R}^N$ as well as a projection map $\mathcal{P} : \mathbb{R}^N \to \mathbb{R}^n$, where $\mathcal{P} \cdot \mathcal{U} = Id_n$. Proposition 3.2 suggests that for quadratic form-based similarity, the choices are $\mathcal{U} = P^+, \mathcal{P} = P$, and $\mathcal{O}_{\widehat{G}} = \widehat{L}$. See the first row in Table 1.

Table 1: Depending on the choice of $\mathcal{F}$ (quantity that we want to preserve) and $\mathcal{O}_G$, we have different projection/lift operators and resulting $\mathcal{O}_{\widehat{G}}$ on the coarse graph.

| Quantity $\mathcal{F}$ of interest | $\mathcal{O}_G$ | Projection $\mathcal{P}$ | Lift $\mathcal{U}$ | $\mathcal{O}_{\widehat{G}}$ | | Invariant under $\mathcal{U}$ |
|---|---|---|---|---|---|---|
| Quadratic form Q | $L$ | $P$ | $P^+$ | Combinatorial Laplace $\widehat{L}$ | | $\mathsf{Q}_L(\mathcal{U}\hat{x}) = \mathsf{Q}_{\widehat{L}}(\hat{x})$ |
| Rayleigh quotient R | $L$ | $\Gamma^{-1/2}(P^+)^T$ | $P^+\Gamma^{-1/2}$ | Doubly-weighted Laplace $\widehat{\mathsf{L}}$ | | $\mathsf{R}_L(\mathcal{U}\hat{x}) = \mathsf{R}_{\widehat{\mathsf{L}}}(\hat{x})$ |
| Quadratic form Q | $\mathcal{L}$ | $\widehat{D}^{1/2}PD^{-1/2}$ | $D^{1/2}(P^+)\widehat{D}^{-1/2}$ | Normalized Laplace $\widehat{\mathcal{L}}$ | | $\mathsf{Q}_{\mathcal{L}}(\mathcal{U}\hat{x}) = \mathsf{Q}_{\widehat{\mathcal{L}}}(\hat{x})$ |

On the other hand, eigenvectors and eigenvalues of a linear operator $A$ are more directly related, via Courant-Fischer Min-Max Theorem, to its Rayleigh quotient $\mathsf{R}_A(x) = \frac{x^T A x}{x^T x}$. Interestingly, in

this case, to preserve the Rayleigh quotient, we should change the choice of $\mathcal{O}_{\widehat{G}}$ to be the following *doubly-weighted Laplace operator* for a graph that is both edge and vertex weighted.

Specifically, for the coarse graph $\widehat{G}$, we assume that each vertex $\hat{v} \in \widehat{V}$ is weighted by $\gamma_{\hat{v}} := |\pi^{-1}(\hat{v})|$, the size of the cluster from $G$ that got collapsed into $\hat{v}$. Let $\Gamma$ be the vertex matrix, which is the $n \times n$ diagonal matrix with $\Gamma[r][r] = \gamma_{\hat{v}_r}$. The *doubly-weighted Laplace operator* for a vertex- and edge-weighted graph $\widehat{G}$ is then defined as:

$$\widehat{\mathsf{L}} = \Gamma^{-1/2}(\widehat{D} - \widehat{W})\Gamma^{-1/2} = \Gamma^{-1/2}\widehat{L}\Gamma^{-1/2} = (P^+\Gamma^{-1/2})^T L (P^+\Gamma^{-1/2}).$$

The concept of doubly-weighted Laplace for a vertex- and edge-weighted graph is not new, see e.g Chung & Langlands (1996); Horak & Jost (2013); Xu et al. (2019). In particular, Horak & Jost (2013) proposes a general form of combinatorial Laplace operator for a simplicial complex where all simplices are weighted, and our doubly-weighted Laplace has the same eigenstructure as their Laplacian when restricted to graphs. See Appendix A.1 for details. Using the doubly-weighted Laplacian for Rayleigh quotient based similarity measurement between the original graph and the coarse graph is justified by the following result (proof in Appendix A.1).

**Proposition 3.3.** *For any vector $x \in \mathbb{R}^n$, we have that $\mathsf{R}_{\widehat{\mathsf{L}}}(\hat{x}) = \mathsf{R}_L(P^+\Gamma^{-1/2}\hat{x})$. That is, set the lift of $\hat{x}$ in $\mathbb{R}^N$ to be $x = P^+\Gamma^{-1/2}\hat{x}$, then we have that $\frac{\hat{x}^T\widehat{\mathsf{L}}\hat{x}}{\hat{x}^T\hat{x}} = \frac{x^T L x}{x^T x}$.*

Finally, if using the normalized Laplace $\mathcal{L}$ for the original graph $G$, then the appropriate Laplace operator for the coarse graph and corresponding projection/lift maps are listed in the last row of Table 1, with proofs in Appendix A.2.

### 3.4 A GNN-BASED FRAMEWORK FOR LEARNING FOR CONSTRUCTING THE COARSE GRAPH

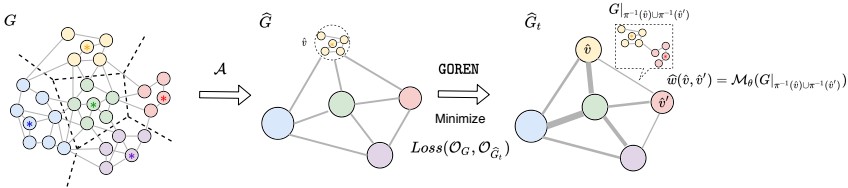

Figure 1: An illustration of learnable coarsening framework. Existing coarsening algorithm determines the topology of coarse graph $\widehat{G}$, while GOREN resets the edge weights of the coarse graph.

In the previous section, we argued that depending on what similarity measures we use, appropriate Laplace operator $\mathcal{O}_{\widehat{G}}$ for the coarse graph $\widehat{G}$ should be used. Now consider the specific case of Rayleigh quotient, which can be thought of as a proxy to measure similarities between the low-frequency eigenvalues of the original graph Laplacian and the one for the coarse graph. As described above, here we set $\mathcal{O}_{\widehat{G}}$ as the doubly-weighted Laplacian $\widehat{\mathsf{L}} = \Gamma^{-1/2}(\widehat{D} - \widehat{W})\Gamma^{-1/2}$.

**The effect of weight adjustments.** We develop an iterative algorithm with convergence guarantee (to KKT point in F.4) for optimizing over edge weights of $\widehat{G}$ for better spectrum alignment. As shown in the figure on the right, after changing the edge weight of the coarse graph, the resulting graph Laplacian has eigenvalues much closer (almost identical) to the first $n$ eigenvalues of the original graph Laplacian. More specifically, in this figure, $G.e$ and $Gc.e$ stand for the eigenvalues of the original graph $G$ and coarse graph $\widehat{G}$ constructed by the so-called Variation-Edge coarsening algorithm (Loukas, 2019). "After-Opt" stands for the eigenvalues of coarse graphs when weights are optimized

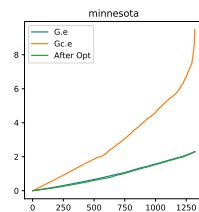

by our iterative algorithm. See Appendix F for the description of our iterative algorithm, its convergence results, and full experiment results.

**A GNN-based framework for learning weight assignment map.** The discussions above indicate that we can obtain better Laplace operators for the coarse graph by using better-informed weights than simply summing up the weights of crossing edges from the two clusters. More specifically, suppose we have a fixed strategy to generate $\widehat{V}$ from an input graph $G = (V, E)$. Now given an edge $(\hat{v}, \hat{v}') \in \widehat{E}$ in the induced coarse graph $\widehat{G} = (\widehat{V}, \widehat{E})$, we model its weight $\hat{w}(\hat{v}, \hat{v}')$ by a

*weight-assignment function* $\mu(G|_{\pi^{-1}(\hat{v}) \cup \pi^{-1}(\hat{v}')})$, where $G|_A$ is the subgraph of $G$ induced by a subset of vertices $A$. However, it is not clear how to setup this function $\mu$. Instead, we will learn it from a collection of input graphs in an *unsupervised* manner. Specifically, we will parametrize the weight-assignment map $\mu$ by a learnable neural network $\mathcal{M}_\theta$. See Figure 1 for an illustration.

In particular, we use Graph Isomorphism Network (GIN) (Xu et al., 2018) to represent $\mathcal{M}_\theta$. We initialize the model by setting the edge attribute of the coarse graph to be 1. Our node feature is set to be a 5-dimensional vector based on LDP (Local Degree Profile) (Cai & Wang, 2018). We enforce the learned weight of the coarse graph to be positive by applying one extra ReLU layer to the final output. All models are trained with Adam optimizer with a learning rate of 0.001. See Appendix E for more details. We name our model as **G**raph c**O**arsening **R**efinem**E**nt **N**etwork (GOREN).

Given a graph $G$ and a coarsening algorithm $\mathcal{A}$, the general form of loss is

$$Loss(\mathcal{O}_G, \mathcal{O}_{\widehat{G}_t}) = \frac{1}{k} \sum_{i=1}^{k} |\mathcal{F}(\mathcal{O}_G, f_i) - \mathcal{F}(\mathcal{O}_{\widehat{G}_t}, \mathcal{P}f_i)|, \tag{1}$$

where $f_i$ is signal on the original graph (such as eigenvectors) and $\mathcal{P}f_i$ is its projection. We use $\mathcal{O}_{\widehat{G}_t}$ to denote the operator of the coarse graph *during training*, while $\mathcal{O}_{\widehat{G}}$ standing for the operator defined w.r.t. the coarse graph output by coarsening algorithm $\mathcal{A}$. That is, we will start with $\mathcal{O}_{\widehat{G}}$ and modify it to $\mathcal{O}_{\widehat{G}_t}$ during the training. The loss can be instantiated for different cases in Table 1. For example, a loss based on quadratic form means that we choose $\mathcal{O}_G, \mathcal{O}_{\widehat{G}_t}$ to be the combinatorial Laplacian of $G$ and $\widehat{G}_t$, and the resulting *quadratic loss* has the form:

$$Loss(L, \widehat{L}_t) = \frac{1}{k} \sum_{i=1}^{k} |f_i^T L f_i - (Pf_i)^T \widehat{L}_t (Pf_i)|. \tag{2}$$

It can be seen as a natural analog of the loss for spectral sparsification in the context of graph coarsening, which is also adopted in Loukas (2019). Similarly, one can use a loss based on the Rayleigh quotient, by choosing $\mathcal{F}$ from the second row of Table 1. Our framework for graph coarsening is flexible. Many different loss functions can be used as long as it is differentiable in the weights of the coarse graph. we will demonstrate this point in Section 4.4.

Finally, given a collection of training graphs $G_1, \ldots, G_m$, we will train for parameters in the module $\mathcal{M}_\theta$ to minimize the total loss on training graphs. When a test graph $G_{test}$ is given, we simply apply $\mathcal{M}_\theta$ to set up weight for each edge in $\widehat{G_{test}}$, obtaining a new graph $\widehat{G_{test,t}}$. We compare $Loss(\mathcal{O}_{G_{test}}, \mathcal{O}_{\widehat{G_{test,t}}})$ against $Loss(\mathcal{O}_{G_{test}}, \mathcal{O}_{\widehat{G_{test}}})$ and expect the former loss is smaller.

## 4 EXPERIMENTS

In the following experiments, we apply six existing coarsening algorithms to obtain the coarsened vertex set $\widehat{V}$, which are Affinity (Livne & Brandt, 2012), Algebraic Distance (Chen & Safro, 2011), Heavy edge matching (Dhillon et al., 2007; Ron et al., 2011), as well as two local variation methods based on edge and neighborhood respectively (Loukas, 2019), and a simple baseline (BL); See Appendix D for detailed descriptions. The two local variation methods are considered to be state-of-the-art graph coarsening algorithms Loukas (2019). We show that our GOREN framework can improve the qualities of coarse graphs produced by these methods.

### 4.1 PROOF OF CONCEPT

As proof of concept, we show that GOREN can improve common coarsening methods on multiple graphs (see C.2 for details). Following the same setting as Loukas (2019), we use the relative eigenvalue error as evaluation metric. It is defined as $\frac{1}{k} \sum_{i=1}^{k} \frac{|\widehat{\lambda}_i - \lambda_i|}{\lambda_i}$, where $\lambda_i, \widehat{\lambda}_i$ denotes eigenvalues of combinatorial Laplacian $L$ for $G$ and doubly-weighted Laplacian $\widehat{L}$ for $\widehat{G}$ respectively, and $k$ is set to be 40. For simplicity,

Table 2: The error reduction after applying GOREN.

| Dataset | Affinity | Algebraic Distance | Heavy Edge | Local var (edges) | Local var (neigh.) |
|---|---|---|---|---|---|
| Airfoil | 91.7% | 88.2% | 86.1% | 43.2% | 73.6% |
| Minnesota | 49.8% | 57.2% | 30.1% | 5.50% | 1.60% |
| Yeast | 49.7% | 51.3% | 37.4% | 27.9% | 21.1% |
| Bunny | 84.7% | 69.1% | 61.2% | 19.3% | 81.6% |

Table 3: Loss: quadratic loss. Laplacian: combinatorial Laplacian for both original and coarse graphs. Each entry $x(y)$ is: $x =$ loss w/o learning, and $y =$ improvement percentage.

| | Dataset | BL | Affinity | Algebraic Distance | Heavy Edge | Local var (edges) | Local var (neigh.) |
|---|---|---|---|---|---|---|---|
| Synthetic | BA | 0.44 (16.1%) | 0.44 (4.4%) | 0.68 (4.3%) | 0.61 (3.6%) | 0.21 (14.1%) | 0.18 (72.7%) |
| | ER | 0.36 (1.1%) | 0.52 (0.8%) | 0.35 (0.4%) | 0.36 (0.2%) | 0.18 (1.2%) | 0.02 (7.4%) |
| | GEO | 0.71 (87.3%) | 0.20 (57.8%) | 0.24 (31.4%) | 0.55 (80.4%) | 0.10 (59.6%) | 0.27 (65.0%) |
| | WS | 0.45 (62.9%) | 0.09 (82.1%) | 0.09 (60.6%) | 0.52 (51.8%) | 0.09 (69.9%) | 0.11 (84.2%) |
| Real | CS | 0.39 (40.0%) | 0.21 (29.8%) | 0.17 (26.4%) | 0.14 (20.9%) | 0.06 (36.9%) | 0.0 (59.0%) |
| | Flickr | 0.25 (10.2%) | 0.25 (5.0%) | 0.19 (6.4%) | 0.26 (5.6%) | 0.11 (11.2%) | 0.07 (21.8%) |
| | Physics | 0.40 (47.4%) | 0.37 (42.4%) | 0.32 (49.7%) | 0.14 (28.0%) | 0.15 (60.3%) | 0.0 (-0.3%) |
| | PubMed | 0.30 (23.4%) | 0.13 (10.5%) | 0.12 (15.9%) | 0.24 (10.8%) | 0.06 (11.8%) | 0.01 (36.4%) |
| | Shape | 0.23 (91.4%) | 0.08 (89.8%) | 0.06 (82.2%) | 0.17 (88.2%) | 0.04 (80.2%) | 0.08 (79.4%) |

this error is denoted as *Eigenerror* in the remainder of the paper. Denote the Eigenerror of graph coarsening method as $l_1$ and Eigenerror obtained by GOREN as $l_2$. In Table 2, we show the *error-reduction ratio*, defined as $\frac{l_1-l_2}{l_1}$. The ratio is upper bounded by 100% in the case of improvement (and the larger the value is, the better); but it is not lower bounded.

Since it is hard to directly optimize Eigenerror, the loss function we use in our GOREN set to be the *Rayleigh loss* $Loss(\mathcal{O}_G, \mathcal{O}_{\widehat{G}_t}) = \frac{1}{k}\sum_{i=1}^{k}|\mathcal{F}(\mathcal{O}_G, f_i) - \mathcal{F}(\mathcal{O}_{\widehat{G}_t}, \mathcal{P}f_i)|$ where $\mathcal{F}$ is Rayleigh quotient, $\mathcal{P} = \Gamma^{-1/2}(P^+)^T$ and $\mathcal{O}_{\widehat{G}_t}$ being doubly-weighted Laplacian $\widehat{\mathsf{L}}_t$. In other words, We use Rayleigh loss as a differentiable proxy for the Eigenerror. As we can see in Table 2, GOREN reduces the Eigenerror by a large margin for *training* graphs, which serves as a sanity check for our framework, as well as for using Rayleigh loss as a proxy for Eigenerror. Due to space limit, see Table G.1 for full results where we reproduce the results in Loukas (2019) up to small differences. In Table 5, we will demonstrate this training strategy also generalizes well to unseen graphs.

## 4.2 SYNTHETIC GRAPHS

We train the GOREN on synthetic graphs from common graph generative models and test on larger unseen graphs from the same model. We randomly sample 25 graphs of size $\{512, 612, 712, ..., 2912\}$ from different generative models. If the graph is disconnected, we keep the largest component. We train GOREN on the first 5 graphs, use the 5 graphs from the rest 20 graphs as the validation set and the remaining 15 as test graphs. We use the following synthetic graphs: Erdős-Rényi graphs (ER), Barabasi-Albert Graph (BA), Watts-Strogatz Graph (WS), random geometric graphs (GEO). See Appendix C.1 for datasets details.

For simplicity, we only report experiment results for the reduction ratio 0.5. For complete results of all reduction ratios (0.3, 0.5, 0.7), see Appendix G. We report both the loss $Loss(L, \widehat{L})$ of different algorithms (w/o learning) and the *relative improvement percentage* defined as $\frac{Loss(L,\widehat{L}) - Loss(L,\widehat{L}_t)}{Loss(L,\widehat{L})}$ when GOREN is applied, shown in parenthesis. As we can see in Table 3, for most methods, trained on small graphs, GOREN also performs well on test graphs of larger size across different algorithms and datasets – Again, the larger improvement percentage is, the larger the improvement by our algorithm is, and a negative value means that our algorithm makes the loss worse. Note the size of test graphs are on average $2.6\times$ the size of training graphs. For ER and BA graphs, the improvement is relatively smaller compared to GEO and WS graphs. This makes sense since ER and BA graphs are rather homogenous graphs, leaving less room for further improvement.

## 4.3 REAL NETWORKS

We test on five real networks: Shape, PubMed, Coauthor-CS (CS), Coauthor-Physics (Physics), and Flickr (largest one with 89k vertices), which are much larger than datasets used in Hermsdorff & Gunderson (2019) ($\leq$ 1.5k) and Loukas (2019) ($\leq$ 4k). Since it is hard to obtain multiple large graphs (except for the Shape dataset, which contains meshes from different surface models) coming from similar distribution, we bootstrap the training data in the following way. For the given graph, we randomly sample a collection of landmark vertices and take a random walk of length $l$ starting from selected vertices. We take subgraphs spanned by vertices of random walks as training and validation graphs and the original graph as the test graph. See Appendix C.3 for dataset details.

Table 4: Loss: quadratic loss. Laplacian: normalized Laplacian for original and coarse graphs. Each entry $x(y)$ is: $x$ = loss w/o learning, and $y$ = improvement percentage.

| | Dataset | BL | Affinity | Algebraic Distance | Heavy Edge | Local var (edges) | Local var (neigh.) |
|---|---|---|---|---|---|---|---|
| Synthetic | BA | 0.13 (76.2%) | 0.14 (45.0%) | 0.15 (51.8%) | 0.15 (46.6%) | 0.14 (55.3%) | 0.06 (57.2%) |
| | ER | 0.10 (82.2%) | 0.10 (83.9%) | 0.09 (79.3%) | 0.09 (78.8%) | 0.06 (64.6%) | 0.06 (75.4%) |
| | GEO | 0.04 (52.8%) | 0.01 (12.4%) | 0.01 (27.0%) | 0.03 (56.3%) | 0.01 (-145.1%) | 0.02 (-9.7%) |
| | WS | 0.05 (83.3%) | 0.01 (-1.7%) | 0.01 (38.6%) | 0.05 (50.3%) | 0.01 (40.9%) | 0.01 (10.8%) |
| Real | CS | 0.08 (58.0%) | 0.06 (37.2%) | 0.04 (12.8%) | 0.05 (41.5%) | 0.02 (16.8%) | 0.01 (50.4%) |
| | Flickr | 0.08 (-31.9%) | 0.06 (-27.6%) | 0.06 (-67.2%) | 0.07 (-73.8%) | 0.02 (-440.1%) | 0.02 (-43.9%) |
| | Physics | 0.07 (47.9%) | 0.06 (40.1%) | 0.04 (17.4%) | 0.04 (61.4%) | 0.02 (-23.3%) | 0.01 (35.6%) |
| | PubMed | 0.05 (47.8%) | 0.05 (35.0%) | 0.05 (41.1%) | 0.12 (46.8%) | 0.03 (-66.4%) | 0.01 (-118.0%) |
| | Shape | 0.02 (84.4%) | 0.01 (67.7%) | 0.01 (58.4%) | 0.02 (87.4%) | 0.0 (13.3%) | 0.01 (43.8%) |

Table 5: Loss: Eigenerror. Laplacian: combinatorial Laplacian for original graphs and doubly-weighted Laplacian for coarse ones. Each entry $x(y)$ is: $x$ = loss w/o learning, and $y$ = improvement percentage. † stands for out of memory.

| | Dataset | BL | Affinity | Algebraic Distance | Heavy Edge | Local var (edges) | Local var (neigh.) |
|---|---|---|---|---|---|---|---|
| Synthetic | BA | 0.36 (7.1%) | 0.17 (8.2%) | 0.22 (6.5%) | 0.22 (4.7%) | 0.11 (21.1%) | 0.17 (-15.9%) |
| | ER | 0.61 (0.5%) | 0.70 (1.0%) | 0.35 (0.6%) | 0.36 (0.2%) | 0.19 (1.2%) | 0.02 (0.8%) |
| | GEO | 1.72 (50.3%) | 0.16 (89.4%) | 0.18 (91.2%) | 0.45 (84.9%) | 0.08 (55.6%) | 0.20 (86.8%) |
| | WS | 1.59 (43.9%) | 0.11 (88.2%) | 0.11 (83.9%) | 0.58 (23.5%) | 0.10 (88.2%) | 0.12 (79.7%) |
| Real | CS | 1.10 (18.0%) | 0.55 (49.8%) | 0.33 (60.6%) | 0.42 (44.5%) | 0.21 (75.2%) | 0.0 (-154.2%) |
| | Flickr | 0.57 (55.7%) | † | 0.33 (20.2%) | 0.31 (55.0%) | 0.11 (67.6%) | 0.07 (60.3%) |
| | Physics | 1.06 (21.7%) | 0.58 (67.1%) | 0.33 (69.5%) | 0.35 (64.6%) | 0.20 (79.0%) | 0.0 (-377.9%) |
| | PubMed | 1.25 (7.1%) | 0.50 (15.5%) | 0.51 (12.3%) | 1.19 (-110.1%) | 0.35 (-8.8%) | 0.02 (60.4%) |
| | Shape | 2.07 (67.7%) | 0.24 (93.3%) | 0.17 (90.9%) | 0.49 (93.0%) | 0.11 (84.2%) | 0.20 (90.7%) |

As shown in the bottom half of Table 3, across all six different algorithms, GOREN significantly improves the result among all five datasets in most cases. For the largest graph Flickr, the size of test graphs is more than $25\times$ of the training graphs, which further demonstrates the strong generalization.

## 4.4 Other losses

**Other differentiable loss.** To demonstrate that our framework is flexible, we adapt GOREN to the following two losses. The two losses are both differentiable w.r.t the weights of coarse graph.

(1) Loss based on normalized graph Laplacian: $Loss(\mathcal{L}, \widehat{\mathcal{L}}_t) = \frac{1}{k} \sum_{i=1}^{k} |f_i^T \mathcal{L} f_i - (\mathcal{P} f_i)^T \widehat{\mathcal{L}}_t (\mathcal{P} f_i)|$. Here $\{f_i\}$ are the set of first $k$ eigenvectors of the normalized Laplacian $\mathcal{L}$ of original grpah $G$, and $\mathcal{P} = \widehat{D}^{1/2} P D^{-1/2}$. (2) Conductance difference between original graph and coarse graph. $Loss = \frac{1}{k} \sum_{i=1}^{k} |\varphi(S_i) - \varphi(\pi(S_i))|$. $\varphi(S)$ is the conductance $\varphi(S) := \frac{\sum_{i \in S, j \in \bar{S}} a_{ij}}{\min(a(S), a(\bar{S}))}$ where $a(S) := \sum_{i \in S} \sum_{j \in V} a_{ij}$. We randomly sample $k$ subsets of nodes $S_0, ..., S_k \subset V$ where $|S_i|$ is set to be a random number sampled from the uniform distribution $U(|V|/4, |V|/2)$. Due to space limits, we present the result for conductance in Appendix G.3.

Following the same setting as before, we perform experiments to minimize two different losses. As shown in Table 4 and Appendix G.3, for most graphs and methods, GOREN still shows good generalization capacity and improvement for both losses. Apart from that, we also observe the initial loss for normalized Laplacian is much smaller than that for standard Laplacian, which might be due to that the fact that eigenvalues of normalized Laplacian are in $[0, 2]$.

**Non-differentiable loss.** In Section 4.1, we use Rayleigh loss as a proxy for training but the Eigenerror for validation and test. Here we train GOREN with Rayleigh loss but evaluate *Eigenerror* on *test* graphs, which is more challenging. Number of vectors $k$ is 40 for synthetic graphs and 200 for real networks. As shown in Table 5, our training strategy via Rayleigh loss can improve the eigenvalue alignment between original graphs and coarse graphs in most cases. Reducing Eigenerror is more challenging than other losses, possibly because we are minimizing a differentiable proxy (the Rayleigh loss). Nevertheless, improvement is achieved in most cases.

Table 6: Model comparison between MLP and GOREN . Loss: quadratic loss. Laplacian: combinatorial Laplacian for both original and coarse graphs. Each entry $x(y)$ is: $x$ = loss w/o learning, and $y$ = improvement percentage.

| Dataset | Ratio | BL | Affinity | Algebraic Distance | Heavy Edge | Local var (edges) | Local var (neigh.) |
|---|---|---|---|---|---|---|---|
| WS + MLP | 0.3 | 0.27 (46.2%) | 0.04 (4.1%) | 0.04 (-38.0%) | 0.43 (31.2%) | 0.02 (-403.3%) | 0.06 (67.0%) |
|  | 0.5 | 0.45 (62.9%) | 0.09 (64.1%) | 0.09 (15.9%) | 0.52 (31.2%) | 0.09 (31.6%) | 0.11 (58.5%) |
|  | 0.7 | 0.65 (70.4%) | 0.15 (57.6%) | 0.14 (31.6%) | 0.67 (76.6%) | 0.15 (43.6%) | 0.16 (54.0%) |
| WS + GOREN | 0.3 | 0.27 (46.2%) | 0.04 (65.6%) | 0.04 (-26.9%) | 0.43 (32.9%) | 0.02 (68.2%) | 0.06 (75.2%) |
|  | 0.5 | 0.45 (62.9%) | 0.09 (82.1%) | 0.09 (60.6%) | 0.52 (51.8%) | 0.09 (69.9%) | 0.11 (84.2%) |
|  | 0.7 | 0.65 (73.4%) | 0.15 (78.4%) | 0.14 (66.7%) | 0.67 (76.6%) | 0.15 (80.8%) | 0.16 (83.2%) |
| Shape + MLP | 0.3 | 0.13 (76.6%) | 0.04 (-53.4%) | 0.03 (-157.0%) | 0.11 (69.3%) | 0.0 (-229.6%) | 0.04 (-7.9%) |
|  | 0.5 | 0.23 (78.4%) | 0.08 (-11.6%) | 0.06 (67.6%) | 0.17 (83.2%) | 0.04 (44.2%) | 0.08 (-1.9%) |
|  | 0.7 | 0.34 (69.9%) | 0.17 (85.1%) | 0.1 (73.5%) | 0.24 (65.8%) | 0.09 (74.3%) | 0.13 (85.1%) |
| Shape + GOREN | 0.3 | 0.13 (86.8%) | 0.04 (79.8%) | 0.03 (69.0%) | 0.11 (69.7%) | 0.0 (1.3%) | 0.04 (73.6%) |
|  | 0.5 | 0.23 (91.4%) | 0.08 (89.8%) | 0.06 (82.2%) | 0.17 (88.2%) | 0.04 (80.2%) | 0.08 (79.4%) |
|  | 0.7 | 0.34 (91.1%) | 0.17 (94.3%) | 0.1 (74.7%) | 0.24 (95.9%) | 0.09 (64.6%) | 0.13 (84.8%) |

## 4.5 ON THE USE OF GNN AS WEIGHT-ASSIGNMENT MAP.

Recall that we use GNN to represent a edge-weight assignment map for an edge $(\hat{u}, \hat{v})$ between two super-nodes $\hat{u}, \hat{v}$ in the coarse graph $\widehat{G}$. The input will be the subgraph $G_{\hat{u},\hat{v}}$ in the original graph $G$ spanning the clusters $\pi^{-1}(\hat{u})$, $\pi^{-1}(\hat{v})$, and the crossing edges among them; while the goal is to compute the weight of edge $(\hat{u}, \hat{v})$ based on this subgraph $G_{\hat{u},\hat{v}}$. Given that the input is a local graph $G_{\hat{u},\hat{v}}$, a GNN will be a natural choice to parameterize this edge-weight assignment map. Nevertheless, in principle, any architecture applicable to graph regression can be used for this purpose. To better understand if it is necessary to use the power of GNN, we replace GNN with the following baseline for graph regression. In particular, the baseline is a composition of mean pooling of node features in the original graph and a 4-layer MLP with embedding dimension 200 and ReLU nonlinearity. We use mean-pooling as the graph regression component needs to be permutation invariant over the set of node features. However, this baseline ignores the detailed graph structure which GNN will leverage. The results for different reduction ratios are presented in the table 6. We have also implemented another baseline where the MLP module is replaced by a simpler linear regression module. The results are worse than those of MLP (and thus also GNN) as expected, and therefore omitted from this paper.

As we can see, MLP works reasonably well in most cases, indicating that learning the edge weights is indeed useful for improvement. On the other hand, we see using GNN to parametrize the map generally yields a larger improvement over the MLP, which ignores the topology of subgraphs in the original graph. A systematic understanding of how different models such as various graph kernels (Kriege et al., 2020; Vishwanathan et al., 2010) and graph neural networks affect the performance is an interesting question that we will leave for future work.

## 5 CONCLUSION

We present a framework to compare original graph and the coarse one via the properly chosen Laplace operators and projection/lift map. Observing the benefits of optimizing over edge weights, we propose a GNN-based framework to learn the edge weights of coarse graph to further improve the existing coarsening algorithms. Through extensive experiments, we demonstrate that our method GOREN significantly improves common graph coarsening methods under different metrics, reduction ratios, graph sizes, and graph types.

## ACKNOWLEDGEMENT

This work is partially supported by National Science Foundation under grants OAC-2039794 and IIS-2050360. Chen Cai would like to thank Huang Fang for helpful discussion on optimization algorithms.

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

## A CHOICE OF LAPLACE OPERATOR

### A.1 LAPLACE OPERATOR ON WEIGHTED SIMPLICIAL COMPLEX

Its most general form in the discrete case, presented as the operators on weighted simplicial complexes, is:

$$\mathcal{L}_i^{up} = W_i^{-1} B_i^T W_{i+1} B_i \quad \mathcal{L}_i^{down} = B_{i-1} W_{i-1}^{-1} B_{i-1}^T W_i$$

where $B_i$ is the matrix corresponding to the coboundary operator $\delta_i$, and $W_i$ is the diagonal matrix representing the weights of $i$-th dimensional simplices. See (Horak & Jost, 2013) for details. When restricted to the graph (1 simplicial complex), we recover the most common graph Laplacians as special case of $\mathcal{L}_0^{up}$. Note that although the $\mathcal{L}_i^{up}$ and $\mathcal{L}_i^{down}$ is not symmetric, we can always symmetrize them by multiple a properly chosen diagonal matrix and its inverse from left and right without altering the spectrum.

### A.2 MISSING PROOFS

We provide the missing proofs regarding the properties of the projection/lift map and the resulting operators on the coarse graph.

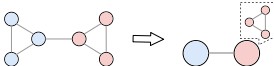

Figure 2: A toy example.

Recall as an toy example, a coarsening algorithm will take graph on the left in figure A.2 and generate a coarse graph on the right,

with coarsening matrix $P = \begin{bmatrix} 1/3 & 1/3 & 1/3 & 0 & 0 & 0 \\ 0 & 0 & 0 & 1/3 & 1/3 & 1/3 \end{bmatrix}$, $P^+ = \begin{bmatrix} 1 & 0 \\ 1 & 0 \\ 1 & 0 \\ 0 & 1 \\ 0 & 1 \\ 0 & 1 \end{bmatrix}$, $\Gamma =$

$\begin{bmatrix} 3 & 0 \\ 0 & 3 \end{bmatrix}$, $\Pi = \begin{bmatrix} 1/3 & 1/3 & 1/3 & 0 & 0 & 0 \\ 1/3 & 1/3 & 1/3 & 0 & 0 & 0 \\ 1/3 & 1/3 & 1/3 & 0 & 0 & 0 \\ 0 & 0 & 0 & 1/3 & 1/3 & 1/3 \\ 0 & 0 & 0 & 1/3 & 1/3 & 1/3 \\ 0 & 0 & 0 & 1/3 & 1/3 & 1/3 \end{bmatrix}$. $\Pi$ in general is a $N \times N$ block matrix of

rank $n$. All entries in each block $\Pi_j$ is equal to $\frac{1}{\gamma_j}$ where $\gamma_j = \left| \pi^{-1}(\hat{v}_j) \right|$.

Table 7: Depending on the choice of $\mathcal{F}$ (quantity that we want to preserve) and $\mathcal{O}_G$, we have different projection/lift operators and resulting $\mathcal{O}_{\widehat{G}}$ on the coarse graph.

| Quantity $\mathcal{F}$ of interest | $\mathcal{O}_G$ | Projection $\mathcal{P}$ | Lift $\mathcal{U}$ | $\mathcal{O}_{\widehat{G}}$ | Invariant under $\mathcal{U}$ |
|---|---|---|---|---|---|
| Quadratic form Q | $L$ | $P$ | $P^+$ | Combinatorial Laplace $\widehat{L}$ | $\mathsf{Q}_L(\mathcal{U}\hat{x}) = \mathsf{Q}_{\widehat{L}}(\hat{x})$ |
| Rayleigh quotient R | $L$ | $\Gamma^{-1/2}(P^+)^T$ | $P^+\Gamma^{-1/2}$ | Doubly-weighted Laplace $\widehat{\mathsf{L}}$ | $\mathsf{R}_L(\mathcal{U}\hat{x}) = \mathsf{R}_{\widehat{\mathsf{L}}}(\hat{x})$ |
| Quadratic form Q | $\mathcal{L}$ | $\widehat{D}^{1/2}PD^{-1/2}$ | $D^{1/2}(P^+)\widehat{D}^{-1/2}$ | Normalized Laplace $\widehat{\mathcal{L}}$ | $\mathsf{Q}_{\mathcal{L}}(\mathcal{U}\hat{x}) = \mathsf{Q}_{\widehat{\mathcal{L}}}(\hat{x})$ |

We first make an observation about projection and lift operator, $\mathcal{P}$ and $\mathcal{U}$.

**Lemma A.1.** $\mathcal{P} \circ \mathcal{U} = I . \mathcal{U} \circ \mathcal{P} = \Pi$.

*Proof.* For the first case, it's easy to see $\mathcal{P} \circ \mathcal{U} = PP^+ = I$ and $\mathcal{U} \circ \mathcal{P} = P^+P = \Pi$.

For the second case, $\mathcal{P} \circ \mathcal{U} = \Gamma^{-1/2}(P^+)^T P^+ \Gamma^{-1/2} = \Gamma^{-1/2}\Pi\Gamma^{-1/2} = I$. $\mathcal{U} \circ \mathcal{P} = P^+\Gamma^{-1}(P^+)^T = I$.

For the third case,

$$\mathcal{P} \circ \mathcal{U} = \widehat{D}^{1/2}PD^{-1/2}D^{1/2}(P^+)\widehat{D}^{-1/2}$$
$$= \widehat{D}^{1/2}P(P^+)\widehat{D}^{-1/2}$$
$$= \widehat{D}^{1/2}I\widehat{D}^{-1/2} = I.$$

$$\mathcal{U} \circ \mathcal{P} = D^{1/2}(P^+)\widehat{D}^{-1/2}\widehat{D}^{1/2}PD^{-1/2}$$
$$= D^{1/2}(P^+)PD^{-1/2}$$
$$= D^{1/2}\Pi D^{-1/2} = \Pi.$$

$\square$

Now we prove the three lemmas in the main paper.

**Proposition A.2.** *For any vector $\hat{x} \in \mathbb{R}^n$, we have that $\mathsf{Q}_{\widehat{L}}(\hat{x}) = \mathsf{Q}_L(P^+\hat{x})$. In other words, set $x := P^+\hat{x}$ as the lift of $\hat{x}$ in $\mathbb{R}^N$, then $\hat{x}^T\widehat{L}\hat{x} = x^T Lx$.*

*Proof.* $\mathsf{Q}_L(\mathcal{U}\hat{x}) = (\mathcal{U}\hat{x})^T L\mathcal{U}\hat{x} = \hat{x}(P^+)^T LP^+\hat{x}^T = \hat{x}^T\widehat{L}\hat{x} = \mathsf{Q}_{\widehat{L}}(\hat{x})$

$\square$

**Proposition A.3.** *For any vector $x \in \mathbb{R}^n$, we have that $R_{\widehat{L}}(\hat{x}) = R_L(P^+ \Gamma^{-1/2} \hat{x})$. That is, set the lift of $\hat{x}$ in $\mathbb{R}^N$ to be $x = P^+ \Gamma^{-1/2} \hat{x}$, then we have that $\frac{\hat{x}^T \widehat{L} \hat{x}}{\hat{x}^T \hat{x}} = \frac{x^T L x}{x^T x}$.*

*Proof.* By definition $R_L(\mathcal{U}\hat{x}) = \frac{Q_L(\mathcal{U}\hat{x})}{||\mathcal{U}\hat{x}||_2^2}$, $R_{\widehat{L}}(x) = \frac{Q_{\widehat{L}}(x)}{||x||_2^2}$. We will prove the lemma by showing $Q_L(\mathcal{U}\hat{x}) = Q_{\widehat{L}}(x)$ and $||\mathcal{U}\hat{x}||_2^2 = ||x||_2^2$.

$$\begin{aligned}
Q_L(\mathcal{U}\hat{x}) &= (\mathcal{U}\hat{x})^T L \mathcal{U}\hat{x} \\
&= \hat{x}^T \Gamma^{-1/2} (P^+)^T L P^+ \Gamma^{-1/2} \hat{x} \\
&= \hat{x}^T \Gamma^{-1/2} \widehat{L} \Gamma^{-1/2} \hat{x} \\
&= \hat{x}^T \widehat{L} \hat{x} \\
&= Q_{\widehat{L}}(\hat{x})
\end{aligned}$$

$||\mathcal{U}\hat{x}||_2^2 = \hat{x}^T \Gamma^{-1/2} (P^+)^T P^+ \Gamma^{-1/2} \hat{x} = \hat{x}^T \hat{x} = ||\hat{x}||_2^2$. Since both numerator and denominator stay the same under the action of $\mathcal{U}$, we conclude $R_L(\mathcal{U}\hat{x}) = R_{\widehat{L}}(\hat{x})$. $\qquad\square$

**Proposition A.4.** *For any vector $x \in \mathbb{R}^n$, we have that $Q_{\widehat{\mathcal{L}}}(x) = Q_{\mathcal{L}}(D^{1/2} P^+ \widehat{D}^{1/2} x)$. That is, set the lift of $\hat{x}$ in $\mathbb{R}^N$ to be $x := D^{1/2} P^+ \widehat{D}^{1/2} x$, then we have that $\hat{x}^T \widehat{\mathcal{L}} \hat{x} = x^T \mathcal{L} x$.*

*Proof.*

$$\begin{aligned}
Q_{\mathcal{L}}(\mathcal{U}\hat{x}) &= (\mathcal{U}\hat{x})^T \mathcal{L} \mathcal{U}\hat{x} \\
&= \hat{x} \widehat{D}^{-1/2} (P^+)^T D^{1/2} \mathcal{L} D^{1/2} (P^+) \widehat{D}^{-1/2} \hat{x} \\
&= \hat{x} \widehat{D}^{-1/2} (P^+)^T L (P^+) \widehat{D}^{-1/2} \hat{x} \\
&= \hat{x} \widehat{D}^{-1/2} \widehat{L} \widehat{D}^{-1/2} \hat{x} \\
&= \hat{x} \widehat{\mathcal{L}} \hat{x} = Q_{\widehat{\mathcal{L}}}(\hat{x})
\end{aligned}$$

$\qquad\square$

# B   MORE RELATED WORK

**Graph pooling**. Graph pooling (Lee et al., 2019) is proposed in the context of the hierarchical graph representation learning. DiffPool (Ying et al., 2018) is proposed to use graph neural networks to parametrize the soft clustering of nodes. Its limitation in quadratic memory is later improved by (Gao & Ji, 2019; Cangea et al., 2018). All those methods are supervised and tested for the graph classification task.

**Optimal transportation theory**. Several recent works adapt the concepts from optimal transportation theory to compare graphs of different sizes. Garg & Jaakkola (2019) aims to minimize optimal transport distance between probability measures on the original graph and coarse graph. Maretic et al. (2019) proposes a framework based on Wasserstein distance between graph signal distributions in terms of their graph Laplacian matrices. Ma & Chen (2019) replaces supervised loss tailored for specific downstream tasks with unsupervised ones based on Wasserstein distance. Dong & Sawin (2020) introduces a novel metric by computing a coordinated pair of optimal transport maps, which is applicable to graph sketching and graph comparison.

# C   DATASET

## C.1   SYNTHETIC GRAPHS

Erdős-Rényi graphs (ER). $G(n, p)$ where $p = \frac{0.1 * 512}{n}$

Random geometric graphs (GEO). The random geometric graph model places $n$ nodes uniformly at random in the unit cube. Two nodes are joined by an edge if the distance between the nodes is at most radius $r$. We set $r = \frac{5.12}{\sqrt{n}}$.

Barabasi-Albert Graph (BA). A graph of $n$ nodes is grown by attaching new nodes each with $m$ edges that are preferentially attached to existing nodes with high degrees. We set $m$ to be 4.

Watts-Strogatz Graph (WS). It is first created from a ring over $n$ nodes. Then each node in the ring is joined to its $k$ nearest neighbors (or $k-1$ neighbors if $k$ is odd). Then shortcuts are created by replacing some edges as follows: for each edge $(u, v)$ in the underlying "$n$-ring with $k$ nearest neighbors" with probability $p$ replace it with a new edge $(u, w)$ with a uniformly random choice of existing node $w$. We set $k, p$ to be 10 and 0.1.

## C.2 Dataset from Loukas's paper

Yeast. Protein-to-protein interaction network in budding yeast, analyzed by (Jeong et al., 2001). The network has $N = 1458$ vertices and $M = 1948$ edges.

Airfoil. Finite-element graph obtained by airow simulation (Preis & Diekmann, 1997), consisting of $N = 4000$ vertices and $M = 11,490$ edges.

Minnesota (Gleich, 2008). Road network with $N = 2642$ vertices and $M = 3304$ edges.

Bunny (Turk & Levoy, 1994). Point cloud consisting of $N = 2503$ vertices and $M = 65,490$ edges. The point cloud has been sub-sampled from its original size.

## C.3 Real networks

Shape graphs (Shape). Each graph is KNN graph formed by 1024 points sampled from shapes from ShapeNet where each node is connected 10 nearest neighbors.

Coauthor-CS (CS) and Coauthor-Physics (Physics) are co-authorship graphs based on the Microsoft Academic Graph from the KDD Cup 2016 challenge. Coauthor CS has $N = 18,333$ nodes and $M = 81,894$ edges. Coauthor Physics has $N = 34,493$ nodes and $M = 247,962$ edges.

PubMed (Sen et al., 2008) has $N = 19,717$ nodes and $M = 44,324$ edges. Nodes are documents and edges are citation links.

Flickr (Zeng et al., 2019) has $N = 89,250$ nodes and $M = 899,756$ edges. One node in the graph represents one image uploaded to Flickr. If two images share some common properties (e.g., same geographic location, same gallery, comments by the same user, etc.), there is an edge between the nodes of these two images.

## D Existing graph coarsening methods

**Heavy Edge Matching**. At each level of the scheme, the contraction family is obtained by computing a maximum-weight matching with the weight of each contraction set $(v_i, v_j)$ calculated as $w_{ij}/\max\{d_i, d_j\}$. In this manner, heavier edges connecting vertices that are well separated from the rest of the graph are contracted first.

**Algebraic Distance**. This method differs from heavy edge matching in that the weight of each candidate set $(v_i, v_j) \in E$ is calculated as $\left(\sum_{q=1}^{Q} (x_q(i) - x_q(j))^2\right)^{1/2}$, where $x_k$ is an $N$-dimensional test vector computed by successive sweeps of Jacobi relaxation. The complete method is described by Ron et al. (2011), see also Chen & Safro (2011).

**Affinity**. This is a vertex proximity heuristic in the spirit of the algebraic distance that was proposed by Livne & Brandt (2012) in the context of their work on the lean algebraic multigrid. As per the author suggests, the $Q = k$ test vectors are here computed by a single sweep of a Gauss-Seidel iteration.

**Local Variation**. There are two variations of local variation methods, edge-based local variation, and neighborhood-based local variation. They differ in how the contraction set is chosen. Edge-based variation is constructed for each edge, while the neighborhood-based variant takes every vertex and its neighbors as contraction set. What two methods have common is that they both optimize an upper bound of the restricted spectral approximation objective. In each step, they greedily pick the sets whose local variation is the smallest. See Loukas (2019) for more details.

**Baseline**. We also implement a simple baseline that randomly chooses a collection of nodes in the original graph as landmarks and contract other nodes to the nearest landmarks. If there are multiple nearest landmarks, we randomly break the tie. The weight of the coarse graph is set to be the sum of the weights of the crossing edges.

## E   DETAILS OF THE EXPERIMENTAL SETUP

**Feature Initialization.**  We initialize the the node feature of subgraphs as a 5 dimensional feature based on a simple heuristics local degree profile (LDP) (Cai & Wang, 2018). For each node $v \in G(V)$, let $DN(v)$ denote the multiset of the degree of all the neighboring nodes of $v$, i.e., $DN(v) = \{\text{degree}(u)|(u,v) \in E\}$. We take five node features, which are (degree($v$), min(DN($v$)), max(DN($v$)),mean(DN($v$)), std(DN($v$))). In other words, each node feature summarizes the degree information of this node and its 1- neighborhood. We use the edge weight as 1 dimensional edge feature.

**Optimization.**  All models are trained with Adam optimizer (Kingma & Ba, 2014) with a learning rate of 0.001 and batch size 600. We use Pytorch (Paszke et al., 2017) and Pytorch Geometric (Fey & Lenssen, 2019) for all of our implementation. We train graphs one by one where for each graph we train the model to minimize the loss for certain epochs (see hyper-parameters for details) before moving to the next graph. We save the model that performs best on the validation graphs and test it on the test graphs.

**Model Architecture.** The building block of our graph neural networks is based on the modification of Graph Isomorphism Network (GIN) that can handle both node and edge features. In particular, we first linear transform both node feature and edge feature to be vectors of the same dimension. At the $k$-th layer, GNNs update node representations by

$$h_v^{(k)} = \text{ReLU}\left(\text{MLP}^{(k)}\left(\sum_{u \in \mathcal{N}(v) \cup \{v\}} h_u^{(k-1)} + \sum_{e=(v,u):u \in \mathcal{N}(v) \cup \{v\}} h_e^{(k-1)}\right)\right) \quad (3)$$

where $\mathcal{N}(v)$ is a set of nodes adjacent to $v$, and $e = (v; v)$ represents the self-loop edge. Edge features $h_e^{(k-1)}$ is the same across the layers.

We use average graph pooling to obtained the graph representation from node embeddings, i.e., $h_G = \text{MEAN}\left(\left\{h_v^{(K)}|v \in G\right\}\right)$. The final prediction of weight is $1 + \text{ReLu}(\Phi(h_G))$ where $\Phi$ is a linear layer. We set the number of layers to be 3 and the embedding dimension to be 50.

**Time Complexity.** In the preprocessing step, we need to compute the first $k$ eigenvectors of Laplacian (either combinatorial or normalized one) of the original graph as test vectors. Those can be efficiently computed by Restarted Lanczos Method (Lehoucq et al., 1998) to find the eigenvalues and eigenvectors.

In the training time, our model needs to recompute the term in the loss involving the coarse graph to update the weights of the graph neural networks for each batch. For loss involving Laplacian (either combinatorial or normalized Laplacian), the time complexity to compute the $x^T L x$ is $O(|E|k)$ where $|E|$ is the number of edges in the coarse graph and $k$ is the number of test vectors. For loss involving conductance, computing the conductance of one subset $S \subset E$ is still $O(|E|)$ so in total the time complexity is also $O(|E|k)$. In summary, the time complexity for each batch is linear in the number of edges of training graphs. All experiments are performed on a single Intel Xeon CPU E5-2630 v4@ 2.20GHz × 40 and 64GB RAM machine.

More concretely, for synthetic graphs, it takes a few minutes to train the model. For real graphs like CS, Physics, PubMed, it takes around 1 hour. For the largest network Flickr of 89k nodes and 899k edges, it takes about 5 hours for most coarsening algorithms and reduction ratios.

**Hyperparameters.** We list the major hyperparameters of GOREN below.

- epoch: 50 for synthetic graphs and 30 for real networks.

- walk length: 5000 for real networks. Note the size of the subgraph is usually around 3500 since the random walk visits some nodes more than once.

- number of eigenvectors $k$: 40 for synthetic graphs and 200 for real networks.

- embedding dimension: 50

- batch size: 600

- learning rate: 0.001

# F  ITERATIVE ALGORITHM FOR SPECTRUM ALIGNMENT

## F.1  PROBLEM STATEMENT

Given a graph $G$ and its coarse graph $\widehat{G}$ output by existing algorithm $\mathcal{A}$, ideally we would like to set edge weight of $\widehat{G}$ so that spectrum of $\widehat{L}$ (denoted as $\mathfrak{L}\mathbf{w}$ below) has prespecified eigenvalues $\boldsymbol{\lambda}$, i.e,

$$\mathfrak{L}\mathbf{w} = U \operatorname{Diag}(\boldsymbol{\lambda}) U^T$$
$$\text{subject to } \mathbf{w} \geq 0, U^T U = I \tag{4}$$

We would like to make an important note that in general, given a sequence of non decreasing numbers $0 = \lambda_1 \leq \lambda_2, ..., \lambda_n$ and a coarse graph $\widehat{G}$, it is not always possible to set the edge weights (always positive) so that the resulting eigenvalues of graph Laplacian of $\widehat{G}$ is $\{0 = \lambda_1, \lambda_2, ..., \lambda_n\}$. We introduce some notations before we present the theorem. The theorem is developed in the context of *inverse eigenvalue problem* for graphs (Barioli & Fallat, 2004; Hogben, 2005; Fallat et al., 2020), which aims to characterize the all possible sets of eigenvalues that can be realized by symmetric matrices whose sparsity pattern is related to the topology of a given graph.

For a symmetric real $n \times n$ matrix $M$, the graph of $M$ is the graph with vertices $\{1, ..., n\}$ and edges $\{\{i, j\} \mid b_{ij} \neq 0 \text{ and } i \neq j\}$. Note that the diagonal of $M$ is ignored in determining $\mathcal{G}(M)$. Let $S_n$ be the set of real symmetric $n \times n$ matrices. For a graph $\widehat{G}$ with $n$ nodes, define $\mathcal{S}(\widehat{G}) = \left\{ M \in S_n \mid \mathcal{G}(M) = \widehat{G} \right\}$.

**Theorem F.1.** *(Barioli & Fallat, 2004; Hogben, 2005) If $T$ is a tree, for any $M \in \mathcal{S}(T)$, the diameter of $T$ is less than the number of distinct eigenvalues of $M$.*

For any graph $\widehat{G}$, its Laplacian (both combinatorial and normalized Laplacian) belongs to $\mathcal{S}(\widehat{G})$, the above theorem therefore applies. In other words, given a tree $T$ and given a sequence of non-decreasing numbers $0 = \lambda_1 \leq \lambda_2, ...\lambda_n$, as long as the number of distinct values in sequences is less than the diameter of $T$, then this sequence can not be realized as the eigenvalues of graph Laplacian of $T$, no matter how we set the edge weights.

Therefore Instead of looking for the a graph with exact spectral alignment with original graph, which is impossible for some nondecreasing sequences as illustrated by the theorem F.1, we relax the equality in equation 4 by instead minimizing the $||\mathfrak{L}\mathbf{w} - U \operatorname{Diag}(\boldsymbol{\lambda}) U^T||_F^2$. We first present an algorithm for the complete graph $\widehat{G}$ of size $n$. This algorithm is essentially the special case of (Kumar et al., 2019). We then show relaxing $\widehat{G}$ from the complete graph to the arbitrary graph will not change the convergence result. Before that, we introduce some notations.

## F.2  NOTATION

**Definition 1.** *The linear operator $\mathfrak{L} : \mathbf{w} \in \mathbb{R}_+^{\frac{n(n-1)}{2}} \to \mathfrak{L}\mathbf{w} \in \mathbb{R}^{n \times n}$ is defined as*

$$[\mathfrak{L}\mathbf{w}]_{ij} = \begin{cases} -w_{i+d_j} & i > j \\ [\mathfrak{L}\mathbf{w}]_{ji} & i > j \\ \sum_{i \neq j}[\mathfrak{L}\mathbf{w}]_{ij} & i = j \end{cases}$$

*where $d_j = -j + \frac{j-1}{2}(2n - j)$*

A toy example is given to illustrate the operators, Consider a weight vector $\mathbf{w} = [w_1, w_2, w_3, w_4, w_5, w_6]^T$, The Laplacian operator $\mathfrak{L}$ on $\mathbf{w}$ gives

$$
\mathfrak{L}\mathbf{w} = \begin{bmatrix} \sum_{i=1,2,3} w_i & -w_1 & -w_2 & -w_3 \\ -w_1 & \sum_{i=1,4,5} w_i & -w_4 & -w_5 \\ -w_2 & -w_4 & \sum_{i=2,4,6} w_i & -w_6 \\ -w_3 & -w_5 & -w_6 & \sum_{i=3,5,6} w_i \end{bmatrix}
$$

Adjoint operator $\mathfrak{L}^*$ is defined to satisfy $\langle \mathfrak{L}\mathbf{w}, Y \rangle = \langle \mathbf{w}, \mathfrak{L}^* Y \rangle$.

### F.3 COMPLETE GRAPH CASE

Recall our goal is to

$$
\begin{aligned}
\underset{\mathbf{w}, U}{\text{minimize}} \quad & \left\| \mathfrak{L}\mathbf{w} - U \operatorname{Diag}(\boldsymbol{\lambda})U^T \right\|_F^2 \\
\text{subject to} \quad & \mathbf{w} \geq 0, U^T U = I
\end{aligned} \tag{5}
$$

---

**Algorithm 1:** Iterative algorithm for edge weight optimization

---

**Input:** coarse graph $\widehat{G}$, error tolerance $\epsilon$, iteration limit $T$
**Output:** coarse graph with new edge weights
1   Initialize $U$ as random element in orthogonal group $O(n, \mathbb{R})$ and $t = 0$.
2   **while** $\epsilon$ *is smaller than the threshold or* $t > T$ **do**
3      Update $\mathbf{w}^{t+1}, U^{t+1}$ according to 8 and Lemma F.4
4      Compute Error $\epsilon$
5      $t = t + 1$
6   From $w^t$, output coarse graph with new edge weights.

---

where $\boldsymbol{\lambda}$ is the desired eigenvalues of the smaller graph. One choice of $\boldsymbol{\lambda}$ can be the first $n$ eigenvalues of the original graph of size $N$. $\mathbf{w}$ and $U$ are variables of size $n(n-1)/2$ and $n \times n$.

The algorithm proceeds by iteratively updating $U$ and $\mathbf{w}$ while fixing the other one.

**Update for w:** It can be seen when $U$ is fixed, minimizing $\mathbf{w}$ is equivalent to a non-negative quadratic problem

$$
\underset{\mathbf{w} \geq 0}{\text{minimize}} \quad f(\mathbf{w}) = \frac{1}{2} \| \mathfrak{L}\mathbf{w} \|_F^2 - \mathbf{c}^T \mathbf{w} \tag{6}
$$

which is strictly convex where $\mathbf{c} = \mathfrak{L}^*(U \operatorname{Diag}(\boldsymbol{\lambda})U^T)$. It is easy to see that the problem is strictly convex. However, due the the non-negativity constraint for $\mathbf{w}$, there is no closed form solution. Thus we derive a majorization function via the following lemma.

**Lemma F.2.** *The function $f(w)$ is majorized at $w_t$ by the function*

$$
g(\mathbf{w}|\mathbf{w}^t) = f(\mathbf{w}^t) + (\mathbf{w} - \mathbf{w}^t)^T \nabla f(\mathbf{w}^t) + \frac{L_1}{2} \left\| \mathbf{w} - \mathbf{w}^t \right\|^2 \tag{7}
$$

*where $\mathbf{w}^t$ is the update from previous iteration an $L_1 = \| \mathfrak{L} \|_2^2 = 2n$.*

After ignoring the constant terms in 7, the majorized problem of 6 at $\mathbf{w}^t$ is given

$$
\underset{\mathbf{w} \geq 0}{\text{minimize}} \quad g(\mathbf{w}|\mathbf{w}^t) = \frac{1}{2}\mathbf{w}^T \mathbf{w} - \boldsymbol{a}^T \mathbf{w}, \tag{8}
$$

where $a = \mathbf{w}^t - \frac{1}{L_1}\nabla f(\mathbf{w}^t)$ and $\nabla f(\mathbf{w}^t) = \mathfrak{L}^*(\mathfrak{L}\mathbf{w}^t) - \mathbf{c}$

**Lemma F.3.** *From the KKT optimality conditions we can easily obtain the optimal solution to 7 as*

$$
\mathbf{w}^{t+1} = (\mathbf{w}^t - \frac{1}{L_1}\nabla f(\mathbf{w}^t))^+
$$

*where $(x)^+ := \max(x, 0)$.*

**Update for $U$:** When $\mathbf{w}$ is fixed, the problem of optimizing $U$ is equivalent to

$$
\begin{aligned}
\underset{U}{\text{minimize}} \quad & \text{tr}(U^T \mathfrak{L}\mathbf{w} U Diag(\boldsymbol{\lambda})) \\
\text{subject to} \quad & U^T U = I
\end{aligned}
\tag{9}
$$

It can be shown that the optimal $U$ at iteration $t$ is achieved by $U^{t+1} = \text{eigenvectors}(L_w)$.

**Lemma F.4.** *From KKT optimality condition, the solution to 9 is given by $U^{t+1} = eigenvectors(\mathfrak{L}\mathbf{w})$.*

The following theorem is proved at (Kumar et al., 2019).

**Theorem F.5.** *The sequence $(\mathbf{w}^t, U^t)$ generated by Algorithm 1 converges to the set of $KKT$ points of 5.*

### F.4 Non-complete Graph Case

The only complication in the case of the non-complete graph is that $\mathbf{w}$ has only $|E|$ number of free variables instead of $\frac{n(n-1)}{2}$ variables as the case of the complete graph. We will argue that $w$ will stay at the subspace of dimension $|E|$ during the iteration.

For simplicity, given a non-compete graph $\widehat{G} = (\widehat{V}, \widehat{E})$, let us denote $\hat{v} = [n] = \{1, 2, ..., n\}$ and each edge will be represented as $(i, j)$ where $i > j$, and $i, j \in [n]$. It is easy to see that we can map each edge $(i, j)$ $(i > j)$ to $k$-th coordinate of $\mathbf{w}$ via $k = \Phi(i, j) = i - j + \frac{(j-1)(2p-j)}{2}$.

Let us denote $\overline{\mathbf{w}}$ (to emphasize its dependence on $\widehat{G}$, it is also denoted as $\mathbf{w}_{\widehat{G}}$ later.) to be the same as $\mathbf{w}$ on coordinates that corresponds to edges in $G$ and 0 for the rest entries. In other words,

$$
\overline{\mathbf{w}}[k] = \begin{cases} \mathbf{w}[k] & \text{if } \Phi^{-1}(k) \in E \\ 0 & \text{o.w.} \end{cases}
$$

Similarly, for any symmetric matrix $A$ of size $n \times n$

$$
\overline{A}[i, j] = \begin{cases} A[i, j] & \text{if}(i, j) \in E \text{ or } (j, i) \in E \\ 0 & \text{o.w.} \end{cases}
$$

Let us also define a $\widehat{G}$-subspace of $\mathbf{w}$ (denoted as $\widehat{G}$-subspace when there is no ambiguity) as $\{\overline{\mathbf{w}} | \mathbf{w} \in \mathbb{R}_+^{n(n-1)/2}\}$. What we need to prove is that if we initialize the algorithm with $\mathbf{w}_{\widehat{G}}$ instead of $\mathbf{w}$, $\mathbf{w}_{\widehat{G}}^t$ will remain in the $\widehat{G}$-subspace of $\mathbf{w}$ for any $t \in \mathbb{Z}_+$.

First, we have the following lemma.

**Lemma F.6.** *We have*

1. $\mathfrak{L}\overline{w} = \overline{\mathfrak{L}w}$.

2. $\langle \overline{\mathbf{w}_1}, \mathbf{w}_2 \rangle = \langle \mathbf{w}_1, \overline{\mathbf{w}_2} \rangle = \langle \overline{\mathbf{w}_1}, \overline{\mathbf{w}_2} \rangle$.

3. $\mathfrak{L}^* \overline{Y} = \overline{\mathfrak{L}^* Y}$

*Proof.* Lemma 1 and 2 can be proved by definition. Now we prove the last lemma. For any $\mathbf{w} \in \mathbb{R}_+^{\frac{n(n-1)}{2}}$ and $Y \in \mathbb{R}^{n \times n}$

$$
\langle \mathbf{w}, \mathfrak{L}^* \overline{Y} \rangle = \langle \mathfrak{L}\mathbf{w}, \overline{Y} \rangle = \langle \overline{\mathfrak{L}\mathbf{w}}, Y \rangle = \langle \mathfrak{L}\overline{\mathbf{w}}, Y \rangle = \langle \overline{\mathbf{w}}, \mathfrak{L}^* Y \rangle = \langle \mathbf{w}, \overline{\mathfrak{L}^* Y} \rangle
$$

where the fourth equation follows from the definition of $\mathfrak{L}^*$ and the other equations directly follows from the previous two lemmas. Therefore $\mathfrak{L}^* \overline{Y} = \overline{\mathfrak{L}^* Y}$. $\square$

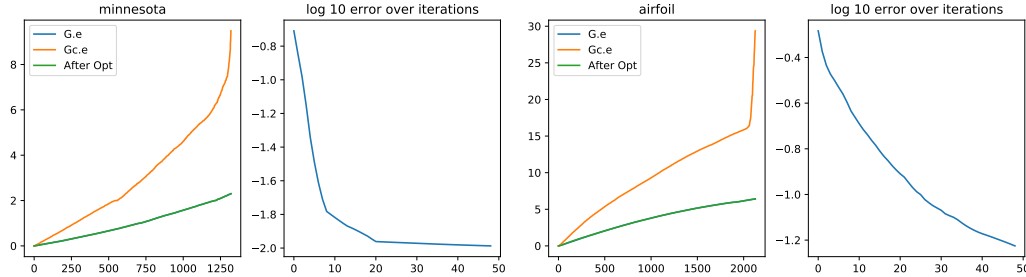

Figure 3: After optimizing edge weights, we can construct a smaller graph with eigenvalues much closer to eigenvalues of original graph. $G.e$ and $Gc.e$ stand for the eigenvalues of original graph and coarse graph output by Variation-Edge algorithm. After-Opt stands for the eigenvalues of graphs where weights are optimized. The error is measured by the maximum absolute difference over *all* eigenvalues of original graph and the coarse graph (after optimization).

Recall that we minimize the following objective when updating $\mathbf{w}_{\widehat{G}}$

$$\underset{\mathbf{w}_{\widehat{G}} \geq 0}{\text{minimize}} \quad \left\| \mathfrak{L}\mathbf{w}_{\widehat{G}} - U \operatorname{Diag}(\boldsymbol{\lambda}) U^T \right\|_F^2$$

which is equivalent to be

$$\underset{\mathbf{w}_{\widehat{G}} \geq 0}{\text{minimize}} \quad \left\| \mathfrak{L}\mathbf{w}_{\widehat{G}} - \overline{U \operatorname{Diag}(\boldsymbol{\lambda}) U^T} \right\|_F^2 \qquad (10)$$

Now following the same process for the case of complete graph. Equation 10 is equivalent to

$$\underset{\mathbf{w}_{\widehat{G}} \geq 0}{\text{minimize}} \quad f(\mathbf{w}_{\widehat{G}}) = \frac{1}{2} \|\mathfrak{L}\mathbf{w}_{\widehat{G}}\|_F^2 - \mathbf{c}^T \mathbf{w}_{\widehat{G}}$$

where $\mathbf{c} = \mathfrak{L}^*(\overline{U \operatorname{Diag}(\boldsymbol{\lambda}) U^T})$.

Use the same majorization function as the case of complete graph, we can get the following update rule

**Lemma F.7.** *From the KKT optimality conditions we can easily obtain the optimal solution to as*

$$\mathbf{w}_{\widehat{G}}^{t+1} = (\mathbf{w}_{\widehat{G}}^t - \frac{1}{L_1} \nabla f(\mathbf{w}_{\widehat{G}}^t))^+$$

*where* $(x)^+ := \max(x, 0)$ *and* $\nabla f(\mathbf{w}_{\widehat{G}}^t) = \mathfrak{L}^*(\mathfrak{L}\mathbf{w}_{\widehat{G}}^t - \overline{U \operatorname{Diag}(\boldsymbol{\lambda}) U^T})$.

Since $\nabla f(\mathbf{w}_{\widehat{G}}^t) = \mathfrak{L}^*(\mathfrak{L}\overline{\mathbf{w}^t}) - \overline{A}) = \mathfrak{L}^*(\overline{\mathfrak{L}\mathbf{w}^t} - \overline{A}) = \overline{\mathfrak{L}^*(\mathfrak{L}\mathbf{w}^t - A)}$ where $A = U \operatorname{Diag}(\boldsymbol{\lambda}) U^T$, therefore $\mathbf{w}_{\widehat{G}}^{t+1}$ will remain in the $\widehat{G}$-subspace if $\mathbf{w}_{\widehat{G}}^t$ is in the $\widehat{G}$-subspace. Since $\mathbf{w}_{\widehat{G}}^0$ is initialized inside $\widehat{G}$-subspace, by induction $\mathbf{w}_{\widehat{G}}^t$ stays in the $\widehat{G}$-subspace for any $t \in \mathbb{Z}^+$. Therefore, we conclude

**Theorem F.8.** *In the case of non-complete graph, the sequence* $(\mathbf{w}^t, U^t)$ *generated by Algorithm 1 converges to the set of KKT points of 5.*

**Remark**: since for each iteration a full eigendecomposition is conducted, the computational complexity is $O(n^3)$ for each iteration, which is certainly prohibitive for large scale application. Another drawback is that the algorithm is not adaptive to the data so we have to run the same algorithm for graphs from the same generative distribution. The main takeaway of this algorithm is that it is possible to improve the spectral alignment of the original graph and coarse graph by optimizing over edge weights, as shown in Figure 3.

Table 8: Relative eigenvalue error (Eigenerror) by different coarsening algorithm and the improvement (in percentage) after applying `GOREN`.

| Dataset | Ratio | Affinity | Algebraic Distance | Heavy Edge | Local var (edges) | Local var (neigh.) |
|---|---|---|---|---|---|---|
| Airfoil | 0.3 | 0.262 (82.1%) | 0.208 (64.9%) | 0.279 (80.3%) | 0.102 (-67.6%) | 0.184 (69.6%) |
| | 0.5 | 0.750 (91.7%) | 0.672 (88.2%) | 0.568 (86.1%) | 0.336 (43.2%) | 0.364 (73.6%) |
| | 0.7 | 2.422 (96.4%) | 2.136 (93.5%) | 1.979 (96.7%) | 0.782 (78.8%) | 0.876 (87.8%) |
| Minnesota | 0.3 | 0.322 (-5.0%) | 0.206 (0.5%) | 0.357 (-4.5%) | 0.118 (-5.9%) | 0.114 (-14.0%) |
| | 0.5 | 1.345 (49.8%) | 1.054 (57.2%) | 0.996 (30.1%) | 0.457 (5.5%) | 0.382 (1.6%) |
| | 0.7 | 4.290 (70.4%) | 3.787 (76.6%) | 3.423 (58.9%) | 2.073 (55.0%) | 1.572 (38.1%) |
| Yeast | 0.3 | 0.202 (10.4%) | 0.108 (5.6%) | 0.291 (1.4%) | 0.113 (6.2%) | 0.024 (-58.3%) |
| | 0.5 | 0.795 (49.7%) | 0.485 (51.3%) | 1.080 (37.4%) | 0.398 (27.9%) | 0.133 (21.1%) |
| | 0.7 | 2.520 (60.4%) | 2.479 (72.4%) | 3.482 (52.9%) | 2.073 (58.9%) | 0.458 (45.9%) |
| Bunny | 0.3 | 0.046 (32.6%) | 0.217 (50.0%) | 0.258 (74.4%) | 0.007 (-328.5%) | 0.082 (74.8%) |
| | 0.5 | 0.085 (84.7%) | 0.372 (69.1%) | 0.420 (61.2%) | 0.057 (19.3%) | 0.169 (81.6%) |
| | 0.7 | 0.182 (84.6%) | 0.574 (78.6%) | 0.533 (75.4%) | 0.094 (45.7%) | 0.283 (73.9%) |

## G    MORE RESULTS

We list the full results from Section 4.4 for loss involving normalized Laplacian and conductance.

### G.1    DETAILS ABOUT SECTION 4.1

We list the full details of Section 4.1.

### G.2    DETAILS ABOUT SECTION 4.2 AND 4.3

We list the full details of Section 4.2 and 4.3.

### G.3    DETAILS ABOUT SECTION 4.4.

We list the full details of Section 4.4.

### G.4    DETAILS ABOUT EIGENERROR

We list the Eigenerror for all datasets when the objective function is $Loss(L, \widehat{L})$.

## H    VISUALIZATION

We visualize the subgraphs corresponding to randomly sampled edges of coarse graphs. For example, in WS graphs, some subgraphs have only a few nodes and edges, while other subgraphs have some common patterns such as the dumbbell shape graph. For PubMed, most subgraphs have tree-like structures, possibly due to the edge sparsity in the citation network.

In Figure H, we visualize the weight difference between coarsening algorithms with and without learning. We also plot the eigenvalues of coarse graphs, where the first 40 eigenvalues of the original graph are smaller than the coarse ones. After optimizing edge weights via `GOREN`, we see both methods produce graphs with eigenvalues closer to the eigenvalues of the original graphs.

Table 9: Loss: quadratic loss. Laplacian: *combinatorial* Laplacian for both original and coarse graphs. Each entry $x(y)$ is: $x$ = loss w/o learning, and $y$ = improvement percentage. BL stands for the baseline.

| Dataset | Ratio | BL | Affinity | Algebraic Distance | Heavy Edge | Local var (edges) | Local var (neigh.) |
|---|---|---|---|---|---|---|---|
| BA | 0.3 | 0.36 (6.8%) | 0.22 (2.9%) | 0.56 (1.9%) | 0.49 (1.7%) | 0.06 (16.6%) | 0.17 (73.1%) |
| | 0.5 | 0.44 (16.1%) | 0.44 (4.4%) | 0.68 (4.3%) | 0.61 (3.6%) | 0.21 (14.1%) | 0.18 (72.7%) |
| | 0.7 | 0.21 (32.0%) | 0.43 (16.5%) | 0.47 (17.7%) | 0.4 (19.3%) | 0.2 (48.2%) | 0.11 (11.1%) |
| CS | 0.3 | 0.25 (28.7%) | 0.08 (24.8%) | 0.05 (21.5%) | 0.09 (15.6%) | 0.0 (-254.3%) | 0.0 (60.6%) |
| | 0.5 | 0.39 (40.0%) | 0.21 (29.8%) | 0.17 (26.4%) | 0.14 (20.9%) | 0.06 (36.9%) | 0.0 (59.0%) |
| | 0.7 | 0.46 (55.5%) | 0.57 (36.8%) | 0.33 (36.6%) | 0.28 (29.3%) | 0.18 (44.2%) | 0.09 (26.5%) |
| Physics | 0.3 | 0.26 (35.4%) | 0.36 (36.6%) | 0.2 (29.7%) | 0.1 (18.6%) | 0.0 (-42.0%) | 0.0 (2.5%) |
| | 0.5 | 0.4 (47.4%) | 0.37 (42.4%) | 0.32 (49.7%) | 0.14 (28.0%) | 0.15 (60.3%) | 0.0 (-0.3%) |
| | 0.7 | 0.47 (60.0%) | 0.53 (55.3%) | 0.42 (61.4%) | 0.27 (34.4%) | 0.25 (67.0%) | 0.01 (-4.9%) |
| Flickr | 0.3 | 0.16 (5.3%) | 0.17 (2.0%) | 0.08 (4.3%) | 0.18 (2.7%) | 0.01 (16.0%) | 0.02 (33.7%) |
| | 0.5 | 0.25 (10.2%) | 0.25 (5.0%) | 0.19 (6.4%) | 0.26 (5.6%) | 0.11 (11.2%) | 0.07 (21.8%) |
| | 0.7 | 0.28 (21.0%) | 0.31 (12.4%) | 0.37 (18.7%) | 0.33 (11.3%) | 0.2 (17.2%) | 0.2 (21.4%) |
| PubMed | 0.3 | 0.17 (13.6%) | 0.06 (6.2%) | 0.03 (9.5%) | 0.1 (4.7%) | 0.01 (18.8%) | 0.0 (39.9%) |
| | 0.5 | 0.3 (23.4%) | 0.13 (10.5%) | 0.12 (15.9%) | 0.24 (10.8%) | 0.06 (11.8%) | 0.01 (36.4%) |
| | 0.7 | 0.31 (41.3%) | 0.23 (22.4%) | 0.14 (8.3%) | 0.14 (-491.6%) | 0.16 (12.5%) | 0.05 (21.2%) |
| ER | 0.3 | 0.25 (0.5%) | 0.41 (0.2%) | 0.2 (0.5%) | 0.23 (0.2%) | 0.01 (4.8%) | 0.01 (5.9%) |
| | 0.5 | 0.36 (1.1%) | 0.52 (0.8%) | 0.35 (0.4%) | 0.36 (0.2%) | 0.18 (1.2%) | 0.02 (7.4%) |
| | 0.7 | 0.39 (3.2%) | 0.55 (2.5%) | 0.44 (2.0%) | 0.43 (0.8%) | 0.23 (2.9%) | 0.29 (10.4%) |
| GEO | 0.3 | 0.44 (86.4%) | 0.11 (65.1%) | 0.12 (81.5%) | 0.34 (80.7%) | 0.01 (0.3%) | 0.14 (70.4%) |
| | 0.5 | 0.71 (87.3%) | 0.2 (57.8%) | 0.24 (31.4%) | 0.55 (80.4%) | 0.1 (59.6%) | 0.27 (65.0%) |
| | 0.7 | 0.96 (83.2%) | 0.4 (55.2%) | 0.33 (54.8%) | 0.72 (90.0%) | 0.19 (72.4%) | 0.41 (61.0%) |
| Shape | 0.3 | 0.13 (86.6%) | 0.04 (79.8%) | 0.03 (69.0%) | 0.11 (69.7%) | 0.0 (1.3%) | 0.04 (73.6%) |
| | 0.5 | 0.23 (91.4%) | 0.08 (89.8%) | 0.06 (82.2%) | 0.17 (88.2%) | 0.04 (80.2%) | 0.08 (79.4%) |
| | 0.7 | 0.34 (91.1%) | 0.17 (94.3%) | 0.1 (74.7%) | 0.24 (95.9%) | 0.09 (64.6%) | 0.13 (84.8%) |
| WS | 0.3 | 0.27 (46.2%) | 0.04 (65.6%) | 0.04 (-26.9%) | 0.43 (32.9%) | 0.02 (68.2%) | 0.06 (75.2%) |
| | 0.5 | 0.45 (62.9%) | 0.09 (82.1%) | 0.09 (60.6%) | 0.52 (51.8%) | 0.09 (69.9%) | 0.11 (84.2%) |
| | 0.7 | 0.65 (73.4%) | 0.15 (78.4%) | 0.14 (66.7%) | 0.67 (76.6%) | 0.15 (80.8%) | 0.16 (83.2%) |

Table 10: Loss: quadratic loss. Laplacian: *normalized* Laplacian for both original and coarse graphs. Each entry $x(y)$ is: $x$ = loss w/o learning, and $y$ = improvement percentage. BL stands for the baseline.

| Dataset | Ratio | BL | Affinity | Algebraic Distance | Heavy Edge | Local var (edges) | Local var (neigh.) |
|---------|-------|-----|----------|-------------------|------------|-------------------|--------------------|
| BA | 0.3 | 0.06 (68.6%) | 0.07 (73.9%) | 0.08 (80.6%) | 0.08 (79.6%) | 0.06 (79.4%) | 0.01 (-15.8%) |
| | 0.5 | 0.13 (76.2%) | 0.14 (45.0%) | 0.15 (51.8%) | 0.15 (46.6%) | 0.14 (55.3%) | 0.06 (57.2%) |
| | 0.7 | 0.22 (17.0%) | 0.23 (5.5%) | 0.24 (10.8%) | 0.24 (9.7%) | 0.23 (5.4%) | 0.17 (36.8%) |
| CS | 0.3 | 0.04 (50.2%) | 0.03 (44.1%) | 0.01 (-7.0%) | 0.03 (50.1%) | 0.0 (-135.0%) | 0.01 (-11.7%) |
| | 0.5 | 0.08 (58.0%) | 0.06 (37.2%) | 0.04 (12.8%) | 0.05 (41.5%) | 0.02 (16.8%) | 0.01 (50.4%) |
| | 0.7 | 0.13 (57.8%) | 0.1 (36.3%) | 0.09 (21.4%) | 0.09 (29.3%) | 0.05 (11.6%) | 0.04 (10.8%) |
| Physics | 0.3 | 0.05 (32.3%) | 0.04 (5.4%) | 0.02 (-16.5%) | 0.03 (69.3%) | 0.0 (-1102.4%) | 0.0 (-59.8%) |
| | 0.5 | 0.07 (47.9%) | 0.06 (40.1%) | 0.04 (17.4%) | 0.04 (61.4%) | 0.02 (-23.3%) | 0.01 (35.6%) |
| | 0.7 | 0.14 (60.8%) | 0.1 (52.0%) | 0.06 (20.9%) | 0.07 (29.9%) | 0.04 (11.9%) | 0.02 (39.1%) |
| Flickr | 0.3 | 0.05 (-29.8%) | 0.05 (-31.7%) | 0.05 (-21.8%) | 0.05 (-66.8%) | 0.0 (-293.4%) | 0.01 (13.4%) |
| | 0.5 | 0.08 (-31.9%) | 0.06 (-27.6%) | 0.06 (-67.2%) | 0.07 (-73.8%) | 0.02 (-440.1%) | 0.02 (-43.9%) |
| | 0.7 | 0.08 (-55.3%) | 0.07 (-32.3%) | 0.04 (-316.0%) | 0.07 (-138.4%) | 0.03 (-384.6%) | 0.04 (-195.6%) |
| PubMed | 0.3 | 0.03 (13.1%) | 0.03 (-15.7%) | 0.01 (-79.9%) | 0.04 (-3.2%) | 0.01 (-191.7%) | 0.0 (-53.7%) |
| | 0.5 | 0.05 (47.8%) | 0.05 (35.0%) | 0.05 (41.1%) | 0.12 (46.8%) | 0.03 (-66.4%) | 0.01 (-118.0%) |
| | 0.7 | 0.09 (58.0%) | 0.09 (34.7%) | 0.07 (68.7%) | 0.07 (21.2%) | 0.08 (67.2%) | 0.03 (43.1%) |
| ER | 0.3 | 0.06 (84.3%) | 0.06 (82.0%) | 0.05 (76.8%) | 0.06 (80.5%) | 0.03 (65.2%) | 0.04 (80.8%) |
| | 0.5 | 0.1 (82.2%) | 0.1 (83.9%) | 0.09 (79.3%) | 0.09 (78.8%) | 0.06 (64.6%) | 0.06 (75.4%) |
| | 0.7 | 0.12 (59.0%) | 0.14 (52.3%) | 0.12 (55.7%) | 0.13 (57.1%) | 0.08 (25.1%) | 0.09 (50.3%) |
| GEO | 0.3 | 0.02 (73.1%) | 0.01 (-37.1%) | 0.01 (-4.9%) | 0.02 (64.8%) | 0.0 (-204.1%) | 0.01 (-22.0%) |
| | 0.5 | 0.04 (52.8%) | 0.01 (12.4%) | 0.01 (27.0%) | 0.03 (56.3%) | 0.01 (-145.1%) | 0.02 (-9.7%) |
| | 0.7 | 0.05 (66.5%) | 0.02 (39.8%) | 0.02 (42.6%) | 0.04 (66.0%) | 0.01 (-56.2%) | 0.02 (0.9%) |
| Shape | 0.3 | 0.01 (82.6%) | 0.0 (41.9%) | 0.0 (25.6%) | 0.01 (87.3%) | 0.0 (-73.6%) | 0.0 (11.8%) |
| | 0.5 | 0.02 (84.4%) | 0.01 (67.7%) | 0.01 (58.4%) | 0.02 (87.4%) | 0.0 (13.3%) | 0.01 (43.8%) |
| | 0.7 | 0.03 (85.2%) | 0.01 (78.9%) | 0.01 (58.2%) | 0.02 (87.9%) | 0.01 (43.6%) | 0.01 (59.4%) |
| WS | 0.3 | 0.03 (78.9%) | 0.0 (-4.4%) | 0.0 (-7.2%) | 0.04 (73.7%) | 0.0 (-253.3%) | 0.01 (60.8%) |
| | 0.5 | 0.05 (83.3%) | 0.01 (-1.7%) | 0.01 (38.6%) | 0.05 (50.3%) | 0.01 (40.9%) | 0.01 (10.8%) |
| | 0.7 | 0.07 (84.1%) | 0.01 (56.4%) | 0.01 (65.7%) | 0.07 (89.5%) | 0.01 (62.6%) | 0.02 (68.6%) |

Table 11: Loss: conductance difference. Each entry $x(y)$ is: $x =$ loss w/o learning, and $y =$ improvement percentage. † stands for out of memory error.

| Dataset | Ratio | BL | Affinity | Algebraic Distance | Heavy Edge | Local var (edges) | Local var (neigh.) |
|---------|-------|-----|----------|--------------------|------------|-------------------|--------------------|
| BA | 0.3 | 0.11 (82.3%) | 0.08 (78.5%) | 0.10 (74.8%) | 0.10 (74.3%) | 0.09 (79.3%) | 0.11 (83.6%) |
| | 0.5 | 0.14 (69.6%) | 0.13 (31.5%) | 0.14 (37.4%) | 0.14 (33.9%) | 0.13 (34.3%) | 0.13 (56.2%) |
| | 0.7 | 0.22 (48.1%) | 0.20 (11.0%) | 0.21 (22.4%) | 0.21 (20.0%) | 0.20 (13.2%) | 0.21 (47.8%) |
| ER | 0.3 | 0.10 (81.0%) | 0.09 (74.7%) | 0.10 (74.3%) | 0.10 (72.5%) | 0.09 (76.4%) | 0.12 (79.0%) |
| | 0.5 | 0.13 (64.0%) | 0.14 (33.8%) | 0.14 (33.6%) | 0.14 (32.5%) | 0.14 (31.9%) | 0.12 (1.4%) |
| | 0.7 | 0.20 (43.4%) | 0.19 (10.1%) | 0.20 (17.6%) | 0.20 (17.2%) | 0.19 (23.4%) | 0.17 (15.7%) |
| GEO | 0.3 | 0.10 (91.2%) | 0.09 (87.0%) | 0.10 (84.8%) | 0.10 (85.5%) | 0.10 (84.6%) | 0.11 (92.3%) |
| | 0.5 | 0.12 (88.1%) | 0.13 (33.9%) | 0.13 (32.6%) | 0.13 (37.6%) | 0.13 (35.3%) | 0.13 (90.1%) |
| | 0.7 | 0.21 (86.7%) | 0.17 (21.9%) | 0.19 (25.2%) | 0.19 (27.3%) | 0.19 (27.8%) | 0.11 (72.4%) |
| Shape | 0.3 | 0.10 (82.3%) | 0.10 (86.8%) | 0.09 (85.8%) | 0.09 (86.3%) | 0.09 (84.8%) | 0.09 (92.0%) |
| | 0.5 | 0.14 (33.2%) | 0.13 (34.7%) | 0.13 (34.6%) | 0.13 (37.7%) | 0.13 (40.8%) | 0.12 (89.8%) |
| | 0.7 | 0.17 (41.4%) | 0.19 (23.4%) | 0.20 (27.7%) | 0.20 (34.0%) | 0.20 (34.3%) | 0.11 (76.8%) |
| WS | 0.3 | 0.10 (86.7%) | 0.09 (82.1%) | 0.10 (84.3%) | 0.10 (82.9%) | 0.09 (81.9%) | 0.10 (90.5%) |
| | 0.5 | 0.13 (80.8%) | 0.13 (31.2%) | 0.13 (33.1%) | 0.13 (27.7%) | 0.13 (34.0%) | 0.13 (86.5%) |
| | 0.7 | 0.19 (45.3%) | 0.19 (19.3%) | 0.19 (27.0%) | 0.19 (26.6%) | 0.20 (27.1%) | 0.11 (12.8%) |
| CS | 0.3 | 0.11 (75.8%) | 0.08 (86.8%) | 0.12 (71.4%) | 0.11 (62.6%) | 0.11 (76.7%) | 0.14 (87.9%) |
| | 0.5 | 0.14 (48.3%) | 0.12 (16.7%) | 0.15 (50.0%) | 0.11 (-7.2%) | 0.11 (6.7%) | 0.09 (9.6%) |
| | 0.7 | 0.26 (40.1%) | 0.22 (29.0%) | 0.24 (35.0%) | 0.24 (41.0%) | 0.23 (35.2%) | 0.17 (28.8%) |
| Physics | 0.3 | 0.10 (81.7%) | 0.07 (79.2%) | 0.11 (73.6%) | 0.10 (73.7%) | 0.11 (79.0%) | 0.13 (4.4%) |
| | 0.5 | 0.13 (20.5%) | 0.19 (39.7%) | 0.15 (27.8%) | 0.16 (31.7%) | 0.15 (25.4%) | 0.11 (-22.3%) |
| | 0.7 | 0.24 (60.2%) | 0.16 (26.1%) | 0.23 (15.3%) | 0.24 (16.5%) | 0.23 (11.2%) | 0.20 (35.9%) |
| PubMed | 0.3 | 0.12 (42.8%) | 0.10 (0.4%) | 0.18 (3.6%) | 0.18 (-0.2%) | 0.19 (0.9%) | 0.11 (26.4%) |
| | 0.5 | 0.15 (19.7%) | 0.19 (1.3%) | 0.24 (-12.9%) | 0.39 (3.7%) | 0.39 (11.8%) | 0.16 (16.0%) |
| | 0.7 | 0.25 (27.3%) | 0.33 (0.8%) | 0.36 (0.0%) | 0.31 (33.2%) | 0.28 (35.3%) | 0.23 (14.1%) |
| Flickr | 0.3 | 0.11 (62.6%) | † | 0.13 (52.5%) | 0.13 (54.7%) | 0.12 (74.2%) | 0.16 (58.3%) |
| | 0.5 | 0.09 (-34.5%) | † | 0.15 (3.1%) | 0.16 (3.4%) | 0.15 (19.9%) | 0.13 (-6.7%) |
| | 0.7 | 0.19 (35.6%) | † | 0.20 (6.0%) | 0.28 (-3.1%) | 0.29 (5.3%) | 0.12 (-25.4%) |

Table 12: Loss: Eigenerror. Laplacian: combinatorial Laplacian for original graphs and *doubly-weighted Laplacian* for coarse graphs. Each entry $x(y)$ is: $x =$ loss w/o learning, and $y =$ improvement percentage. † stands for out of memory error.

| Dataset | Ratio | BL | Affinity | Algebraic Distance | Heavy Edge | Local var (edges) | Local var (neigh.) |
|---|---|---|---|---|---|---|---|
| BA | 0.3 | 0.19 (4.1%) | 0.1 (5.4%) | 0.12 (5.6%) | 0.12 (5.0%) | 0.03 (25.4%) | 0.1 (-32.2%) |
| | 0.5 | 0.36 (7.1%) | 0.17 (8.2%) | 0.22 (6.5%) | 0.22 (4.7%) | 0.11 (21.1%) | 0.17 (-15.9%) |
| | 0.7 | 0.55 (9.2%) | 0.32 (12.4%) | 0.39 (10.2%) | 0.37 (10.9%) | 0.21 (33.0%) | 0.28 (-29.5%) |
| CS | 0.3 | 0.46 (16.5%) | 0.3 (56.9%) | 0.11 (59.1%) | 0.23 (38.9%) | 0.0 (-347.6%) | 0.0 (-191.8%) |
| | 0.5 | 1.1 (18.0%) | 0.55 (49.8%) | 0.33 (60.6%) | 0.42 (44.5%) | 0.21 (75.2%) | 0.0 (-154.2%) |
| | 0.7 | 2.28 (16.9%) | 0.82 (57.0%) | 0.66 (53.3%) | 0.73 (38.9%) | 0.49 (73.4%) | 0.34 (63.3%) |
| Physics | 0.3 | 0.48 (19.5%) | 0.35 (67.2%) | 0.14 (65.2%) | 0.2 (57.4%) | 0.0 (-521.6%) | 0.0 (20.7%) |
| | 0.5 | 1.06 (21.7%) | 0.58 (67.1%) | 0.33 (69.5%) | 0.35 (64.6%) | 0.2 (79.0%) | 0.0 (-377.9%) |
| | 0.7 | 2.11 (19.1%) | 0.88 (72.9%) | 0.62 (66.7%) | 0.62 (64.9%) | 0.31 (70.3%) | 0.01 (-434.0%) |
| Flickr | 0.3 | 0.33 (20.4%) | † | 0.16 (7.8%) | 0.16 (9.1%) | 0.02 (63.0%) | 0.04 (-88.9%) |
| | 0.5 | 0.57 (55.7%) | † | 0.33 (20.2%) | 0.31 (55.0%) | 0.11 (67.6%) | 0.07 (60.3%) |
| | 0.7 | 0.86 (85.2%) | † | 0.6 (32.6%) | 0.57 (38.7%) | 0.23 (92.2%) | 0.21 (40.7%) |
| PubMed | 0.3 | 0.56 (5.6%) | 0.27 (13.8%) | 0.13 (17.4%) | 0.34 (10.6%) | 0.06 (-0.4%) | 0.0 (31.1%) |
| | 0.5 | 1.25 (7.1%) | 0.5 (15.5%) | 0.51 (12.3%) | 1.19 (-110.1%) | 0.35 (-8.8%) | 0.02 (60.4%) |
| | 0.7 | 2.61 (8.9%) | 1.12 (19.4%) | 2.24 (-149.8%) | 4.31 (-238.6%) | 1.51 (-260.2%) | 0.27 (75.8%) |
| ER | 0.3 | 0.27 (-0.1%) | 0.35 (0.4%) | 0.15 (0.6%) | 0.18 (0.5%) | 0.01 (5.7%) | 0.01 (-10.4%) |
| | 0.5 | 0.61 (0.5%) | 0.7 (1.0%) | 0.35 (0.6%) | 0.36 (0.2%) | 0.19 (1.2%) | 0.02 (0.8%) |
| | 0.7 | 1.42 (0.8%) | 1.27 (2.1%) | 0.7 (1.4%) | 0.68 (0.3%) | 0.29 (3.5%) | 0.33 (10.2%) |
| GEO | 0.3 | 0.78 (43.4%) | 0.08 (80.3%) | 0.09 (77.1%) | 0.27 (82.2%) | 0.01 (-524.6%) | 0.1 (82.5%) |
| | 0.5 | 1.72 (50.3%) | 0.16 (89.4%) | 0.18 (91.2%) | 0.45 (84.9%) | 0.08 (55.6%) | 0.2 (86.8%) |
| | 0.7 | 3.64 (30.4%) | 0.33 (86.0%) | 0.25 (86.7%) | 0.61 (93.0%) | 0.15 (88.7%) | 0.32 (79.3%) |
| Shape | 0.3 | 0.87 (55.4%) | 0.12 (88.6%) | 0.07 (56.7%) | 0.29 (80.4%) | 0.01 (33.1%) | 0.09 (84.5%) |
| | 0.5 | 2.07 (67.7%) | 0.24 (93.3%) | 0.17 (90.9%) | 0.49 (93.0%) | 0.11 (84.2%) | 0.2 (90.7%) |
| | 0.7 | 4.93 (69.1%) | 0.47 (94.9%) | 0.27 (68.5%) | 0.71 (95.7%) | 0.25 (79.1%) | 0.34 (87.4%) |
| WS | 0.3 | 0.7 (32.3%) | 0.05 (84.7%) | 0.04 (58.9%) | 0.44 (37.3%) | 0.02 (75.0%) | 0.06 (83.4%) |
| | 0.5 | 1.59 (43.9%) | 0.11 (88.2%) | 0.11 (83.9%) | 0.58 (23.5%) | 0.1 (88.2%) | 0.12 (79.7%) |
| | 0.7 | 3.52 (45.6%) | 0.18 (77.7%) | 0.17 (78.2%) | 0.79 (82.8%) | 0.17 (90.9%) | 0.19 (65.8%) |

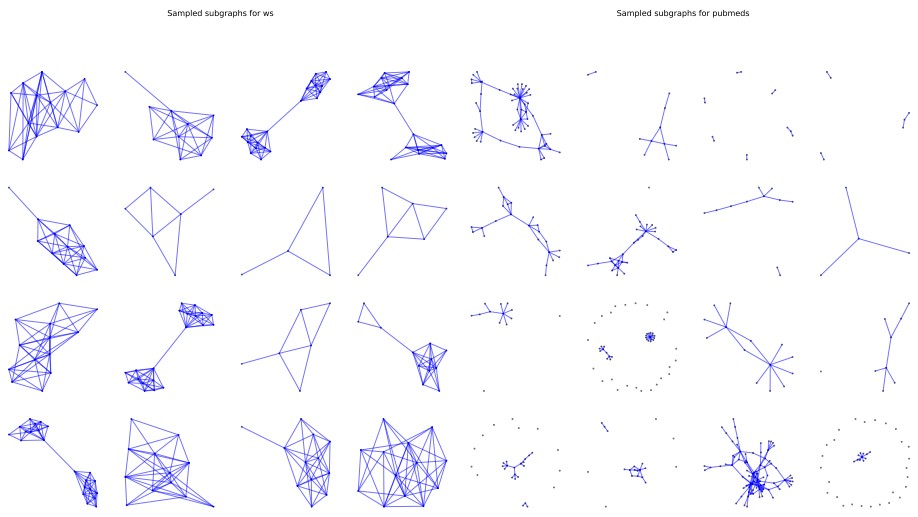

Figure 4: A collection of subgraphs corresponding to edges in coarse graphs (WS and PubMed) generated by variation neighborhood algorithm. Reduction ratio is 0.7 and 0.9 respectively.

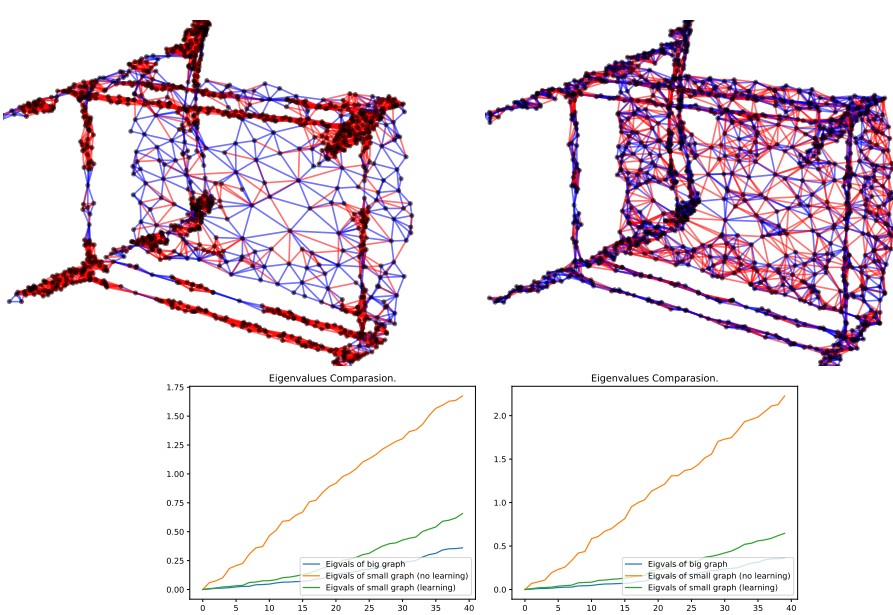

Figure 5: The first row illustrates the weight difference for local variation neighborhood (left) and heavy edge (right) with 0.5 as the reduction ratio. Blue (red) edges denote edges whose learned weights is smaller (larger) than the default ones. The second row shows the first 40 eigenvalues of the original graph Laplacian, coarse graph w/o learning, and coarse graph w/ learning.

