# OpenReview forum: "Graph Coarsening with Neural Networks"
_ICLR.cc/2021/Conference — ICLR 2021 Poster_

### Official Review · AnonReviewer1 · 2020-10-28
**A ML framework for learning coarsen graph edge weights**

**Rating:** 6
**Confidence:** 4

**Review:**

Comment before review: This submission seems to use a different margin. The margins of tables and figures are also tiny which makes the submission hard to read. For example, see Table 1. UPDATE: seems like the authors have corrected the margin issue.

Review:

This submission proposes a machine learning framework to learn the edge weights of coarsened graphs. The authors propose to use graph neural network (specifically, Graph Isomorphism Network) to embed the nodes being coarsened and use the sub-graph embeddings to compute coarsened graph edges. The proposed method leads to a reduction of the eigen error of the coarsened graphs. As far as I know, the proposed method is novel.

Comment on writing:
I think the current presentation complicates the introduction to the proposed method. The description of the proposed method does not appear until page 5 (aside from the introduction). I recommend the authors to condense the discussion in section 3.1 through section 3.3 and focus more on their own contribution

Strength of the submission:
- the proposed method is novel
- the proposed method demonstrates empirical improvement

Weakness of the submission:
- My main concern with this submission is that it lacks an understanding of the proposed method. This paper points out that learning-based method can further reduce the eigenerror by assigning better weights to the coarsened graph, which I am convinced. However, I am less convinced about the proposed learning process. Is the use of a graph neural network really necessary? Considering the input to the GNN here is just simple degree statistics, I doubt if the network can learn much. The author should consider using linear model or simpler models like MLP to learn the edge weights. Without a comprehensive comparison to these baseline, I would not be convinced.

Typos:
- In abstract, "adaptive to different loss" -> "is adaptive / adapts to different loss"


Updated review:

I thank the author for their new experiments during the discussion period. Given the superior performance of GNN over MLP, I am more convinced that the usage of GNNs in this application is justified. I have updated my review rating from 4 -> 6 to reflect this.

But just to harass the authors a bit more, I have this curious question:
- Is the worse performance of MLP due to generalization or expressive power? In other words, can the MLP fit the training data well? Also, when comparing MLP and GOREN, are the authors controlling the number of parameters when comparing MLP and GOREN? As the authors mentioned in their reply, which I agree, "MLP generally works better than LR due to model capacity". We want to make sure that MLP and GOREN have similar model capacity but GOREN captures better inductive biases.

I believe this submission finds an interesting application for GNNs. I encourage the authors to bring out the full potential of this idea by having solid, rigorous empirical studies.

---

> ### Author Response · Authors · 2020-11-10
> **Clarification**
>
> Thank you for your comments and feedback! We will be responding to your and all other reviews  (and our thanks to all the reviewers) soon. But first we hope to get clarification on one of your comments.
>
> We are wondering whether you could clarify what you mean by a linear (or other) model as a baseline to learn the edge weight in your main concern? Note that we are learning the edge weight for edge (u, v) based on the local neighborhoods of both super-node u and super-node v. Such a neighborhood also contains the cluster of nodes from the original graph collapsed (mapped) to each supernode u and v, as well as the crossing edges among the two corresponding clusters. This neighborhood is naturally modeled as a small local graph, and that is why to have a learnable component that can take in such a neighborhood and predict a value (edge weight), GNN is the most natural choice.
>
> In particular, note that this neighborhood is unstructured and of varying size for different edges -- it is not clear how to have a simple linear model or MLP directly on such input. One alternative way we see is to model this neighborhood as a set and use DeepSet-like architecture (as the set is permutation invariant). However, note that it will then ignore the local connections, using only the collection of node features or edge features, and also it is not clear that DeepSet is much simpler than GNN either.
>
> Another way is to use the node feature in the coarse graph and build an MLP based on it. In particular, for two supernodes u, v in the coarse graph, we build an MLP to map from f_u + f_v (or sth. similar) to the edge weight, where f_* is the node feature for node *. This baseline doesn’t utilize the neighborhoods at all, not the set of original graph nodes mapped to these super-nodes. Thus we think that it may not be informative to predict good weight for edges among super nodes. Is this what you have in mind? We would be happy to implement and compare with such a baseline if you could help to clarify.

---

> > ### Comment · AnonReviewer1 · 2020-11-11
> > **Response**
> >
> > Thank you for a quick response.
> >
> > Again, I want to stress that my main concern with the experiments is that there is no comparison to any other (learning-based) baseline.
> >
> > Here are some suggestions for finding baselines that can hopefully convince me:
> > - Clarifying the MLP suggestion: in my understanding, for graph classification, the standard way to use MLP is to transform each node's features and apply pooling (max, mean, sum). If MLP is a reasonable baseline for graph classification ([1]), I believe it can also be applied here.
> > - I thinking the last baseline suggested in the response is a good starting point. If a MLP built on the coarse graph node features can perform well, then this suggests that the GNN is not doing what we thought it is doing. Therefore, it would be nice to sanity check against it.
> > - Can more classic methods for graph classification be useful here? Based on the results in the GIN paper [2], WL subtree kernel method seems to be a strong candidate. Can we use the subtree features of the original local graph and learn a regressor on top of it? I am not an expert with the subtree kernel method but if it can work for graph classification, intuitively it can be applied to this problem?
> > - other ablation studies can also be helpful: even if using GNN is necessary, is it important to consider both the neighborhood in the coarsened graph and the original graph? Will either one of them suffice? This provides insights into what features the GNN is relying on.
> >
> > Overall, I think this submission finds a very interesting application for graph learning methods. If the authors can improve the analysis of the proposed method, this paper should be valuable to the larger community. My understanding is that the empirical experiments should not take long. If within the discussion period, the author can compare to 1 or 2 reasonable baselines on 1 or 2 datasets they studied and show that the use of GNN is valid, then I am happy to be convinced and change my review opinion. Otherwise, I would suggest to include other learning methods into the paper as well.
> >
> > [1] Open Graph Benchmark: Datasets for Machine Learning on Graphs. Weihua Hu and Matthias Fey and Marinka Zitnik and Yuxiao Dong and Hongyu Ren and Bowen Liu and Michele Catasta and Jure Leskovec. arxiv preprint, arxiv: 2005.00687.
> > [2] How Powerful are Graph Neural Networks? Keyulu Xu and Weihua Hu and Jure Leskovec and Stefanie Jegelka. International Conference on Learning Representations, 2019.

---

> > > ### Author Response · Authors · 2020-11-17
> > > **Learning based baselines**
> > >
> > > Thank you for your review, and thank you very much for the clarification in the response above.
> > >
> > > First, we would like to apologize for the margin issue in the original submission. It was unintentional -- we copied some macros from our other papers, which included commands that adjusted the margins. It was fixed once it was brought to our attention.
> > >
> > > Regarding your question on a more informed baseline for the edge-assignment map, thank you for the suggestion. To summarize: we have performed new experiments with two new baselines (one linear regression based, one MLP based), and have also added results/discussion in a new subsection (Section 4.5) in the revised submission. (The results using linear regression are not reported in the revision, but are included below.)
> > >
> > > More specifically, recall that we use GNN to represent an edge-weight assignment map for an edge $(\hat{u}$, $\hat{v}$) between two super-nodes $\hat{u}$ and $\hat{v}$ in the coarse graph $\widehat{G}$. The input will be the subgraph $G_{\hat{u},\hat{v}}$ in the original graph $G$ spanning the clusters of nodes $\pi^{-1}(\hat{u})$, $\pi^{-1}(\hat{v})$ (i.e, the set of original nodes mapped to $\hat{u}$ and to $\hat{v}$, respectively), and the crossing edges among them. The goal is to compute the weight of edge ($\hat{u}$, $\hat{v}$) based on this subgraph $G_{\hat{u}, \hat{v}}$.
> > >
> > > Given that the input is a local graph $G_{\hat{u}, \hat{v}}$, a GNN is perhaps a most natural choice to parameterize this edge-weight assignment map. Nevertheless, in principle, any architecture applicable to graph regression can be used for this purpose.
> > > To better understand if it is necessary to use the power of GNN, we replace GNN with the following two baselines for graph regression. The first baseline is the composition of mean pooling of node features in the original graph and linear regression (LR). The second baseline is a composition of mean pooling of node features in the original graph and a 4-layer MLP with embedding dimension 50 and ReLU nonlinearity. Note that while MLP is more sophisticated than LR, both baselines ignore the detailed graph structure which our GNN will leverage. The results for reduction ratio of 0.5 are shown below. See table 6 in the revised paper for full results (results on LR are not included in the revised manuscript as they are generally worse than those of MLP).
> > >
> > >  BL 	Affinity                  Algebraic                                 Heavy-edge        Local-var(edges)    Local-var (nbrs)
> > >
> > > WS + LR         0.45 (42.9%)   	0.09 (51.9)         0.09 (1.3%)             0.52 (10.3%)        0.09 (28.7%)         0.11 (52.2%)
> > >
> > > WS + MLP      0.45 (62.9%)	0.09 (64.1%)      0.09 (15.9%)           0.52 (31.2%)       0.09 (31.6%)           0.11 (58.5%)
> > >
> > > WS + GOREN 0.45 (62.9%)	0.09 (82.1%)      0.09 (60.6%)           0.52 (51.8%)        0.09 (69.9%)          0.11 (84.2%)
> > >
> > >
> > > Shape + LR        0.23 (80.4%)  0.08 (-11.6%)   0.06 (-71.3%)               0.17 (78.2%)      0.04 (-263.3%)         0.08 (-1.9%)
> > >
> > > Shape + MLP      0.23 (78.4%) 0.08 (-11.6%)     0.06 (67.6%)           0.17 (83.2%)       0.04 (44.2%)           0.08 (-1.9%)
> > >
> > > Shape + GOREN  0.23(91.4%) 0.08 (89.8%)      0.06 (82.2%)           0.17 (88.2%)        0.04 (80.2%)          0.08 (79.4%)
> > >
> > > As we can see, LR/MLP works reasonably well in most cases, indicating that learning the edge weights is crucial for improvement. But MLP generally works better than LR due to model capacity. On the other hand, we see using GNN to parametrize the map generally yields a larger improvement over the MLP, as MLP does not utilize the topology of subgraph while GNN does. A systematic understanding of how different models such as various graph kernels [1, 2] and graph neural networks affect the performance is an interesting question that we will leave for future work.
> > >
> > > [1] Vishwanathan, S. Vichy N., et al. "Graph kernels." The Journal of Machine Learning Research 11 (2010): 1201-1242.
> > >
> > > [2] Kriege, Nils M., Fredrik D. Johansson, and Christopher Morris. "A survey on graph kernels." Applied Network Science 5.1 (2020): 1-42.

---

> > > > ### Comment · AnonReviewer1 · 2020-11-17
> > > > **Repeating my question in updated review**
> > > >
> > > > - Is the worse performance of MLP due to generalization or expressive power? In other words, can the MLP fit the training data well? Also, when comparing MLP and GOREN, are the authors controlling the number of parameters when comparing MLP and GOREN? As the authors mentioned in their reply, which I agree, "MLP generally works better than LR due to model capacity". We want to make sure that MLP and GOREN have similar model capacity but GOREN captures better inductive biases

---

> > > > > ### Author Response · Authors · 2020-11-24
> > > > > **training error**
> > > > >
> > > > > Thank you for your question. Indeed, we controlled the number of parameters: In fact, the number of parameters of MLP is roughly 30 percent larger than that of GOREN. As far as training error is concerned, the MLP and GOREN have similar training errors. In the table below, we list the training error and the improvement ratio (in parenthesis) in each entry; so for example, "0.03 (62.9%)" means that the training error is 0.03, while the improvement over testing datasets is 62.9% (this number is the larger the better). All these indicate that including graph convolutional layers (that leverage the topology of subgraphs) is crucial to achieving better generalization in our GOREN framework.
> > > > >
> > > > > BL     	Affinity                  Algebraic                                 Heavy-edge        Local-var(edges)    Local-var (nbrs)
> > > > >
> > > > > WS + MLP      0.03(62.9%)  0.04 (64.1%)      0.03 (15.9%)       	0.03 (31.2%)   	   0.03 (31.6%)    	0.06 (58.5%)
> > > > >
> > > > > WS + GOREN 0.03(62.9%)  0.04 (82.1%)     0.04 (60.6%)        	0.03 (51.8%)   	  0.04 (69.9%)    	0.08 (84.2%)
> > > > >
> > > > > Shape + MLP   0.02(78.4%)  0.02 (-11.6%)	 0.02 (67.6%)      	0.02 (83.2%)   	  0.01 (44.2%)    	0.01 (-1.9%)
> > > > >
> > > > > Shape + GOREN  0.02(91.4%)0.01 (89.8%)  	0.02 (82.2%)           0.02 (88.2%)        0.01 (80.2%)         0.01 (79.4%)

---

### Official Review · AnonReviewer4 · 2020-10-28
**Solid paper with theoretical and experimental results.**

**Rating:** 6
**Confidence:** 3

**Review:**

The paper studied the problem of graph coarsening in the context of data-driven deep generative models. The authors studied a family of Laplacian operators and differentiable losses in order to construct high-quality coarse graphs. The paper is well-motivated by providing extensive theoretical analysis to support the rationale of the proposed model. The proposed model is developed based on GAN, which automatically learns a mapping function from the fine-grained graph to the coarse-grained graph. Experimental results show the effectiveness of the proposed model across a bunch of datasets (both synthetic and real ones) and a set of evaluation metrics.

In general, I believe this paper is well-written, and the results are strong.
My only concern comes from the technical contribution of the proposed algorithm. In particular, the authors claimed that " we are the first to propose and develop a framework to learn coarse graphs with GNN in an unsupervised manner". However, similar ideas (e.g., Misc-GAN) have already been approached in the network generation setting.  Although, in the graph generation setting, the previous work constructs coarse graphs for the purpose of preserving hierarchical network structures, while in the graph coursing setting, the goal is to alleviate the computational challenges in dealing with massive graphs. But, regarding the framework design, they share some commonalities at a high-level (i.e., GAN-based models for constructing coarse graphs). Please correct me if I am wrong here.

Without that, I have no question regarding this paper. Moreover, if the author can clear my only concerns above, I would like to higher my score to 7.

---

> ### Author Response · Authors · 2020-11-17
> **Responses to Reviewer4**
>
> Thank you very much for your comments and feedback!
>
> Thank you for the reference of misc-GAN. Indeed, there is certain high-level similarity in the sense that both use coarse graphs, and both are unsupervised methods. Misc-GAN is a GAN based deep generative model which constructs coarse graphs via the algebraic multigrid method for the purpose of preserving hierarchical network structures. However, the coarsened graphs themselves are not learned. While in our approach, we propose to learn a coarsened graph in the sense of learning the edge-weight assignment map, which is parameterized by a GNN. To our best knowledge, learning such a coarsened graph in an unsupervised manner has not been considered before. Nevertheless, we think this is a good point -- We have now added a paragraph in the related work section on the deep generative models for graphs, including the misc-GAN reference [1].
>
>
> [1] Zhou, Dawei, et al. "Misc-GAN: A Multi-scale Generative Model for Graphs." Frontiers in Big Data 2 (2019): 3.

---

### Official Review · AnonReviewer3 · 2020-10-29
**Graph Coarsening with Neural Networks**

**Rating:** 7
**Confidence:** 3

**Review:**

Summary of the paper: To solve the problem of graph coarsening, this paper proposes a data-driven framework to: 1) measure the quality of the coarsening algorithm and 2) provide a graph neural network (GNN)-based method to handle the suboptimal problem of edge weight occurring in current methods. The proposed model can handle larger graphs than previous methods. The experimental results demonstrate the effectiveness of the proposed method.

Recommendation: I think contributions of this paper on graph coarsening are new and technically solid. The authors propose a new way (GNN model) to do graph coarsening. Some strong points: (1) The authors provide three new projection operators on graph coarsening. (2) A new framework is proposed to learn better edge weights of the coarse graphs by using a GNN model in an unsupervised manner. (3) Empirical results support their findings (most results are significantly better than baseline methods). Based on these observations, I tend to accept this paper.

---

> ### Author Response · Authors · 2020-11-17
> **Thank you!**
>
> Thank you very much for your positive feedback!

---

### Official Review · AnonReviewer2 · 2020-10-29
**Nice formulations with reasonable approaches**

**Rating:** 7
**Confidence:** 4

**Review:**

This paper studies a graph coarsening strategy where a new way of assigning weights to a coarse graph is proposed. By focusing on preserving properties of the Laplace operator, appropriate projection/lift operators are presented. Based on the observation that better-informed weights enable us to obtain better Laplace operators for coarse graph, a GNN-based weight adjustment method is proposed. The proposed method called GOREN learns the weight-assignment map $\mu$ from a collection of input graphs in an unsupervised manner, and can be generalized to test graphs of larger size than the training graphs. Experimental results show that GOREN improves common graph coarsening methods under different evaluation metrics.

Overall, the paper is well-written, and the contributions are concrete. The Laplace operator is considered to be one of the most important operators because it can explain much about the graph structure. When a graph is converted to a coarse graph, preserving the properties of the Laplace operator can be a critical issue. This paper nicely formulates this issue and the theoretical analysis seems to be technically sound (I did not thoroughly check all the details, though).

In terms of the graph reduction ratio, the values of (0.3, 0.5, 0.7) are chosen in the experiments and the authors simply enumerate the results according to the reduction ratios. I'm wondering if there is a way to find an appropriate reduction ratio by theoretical analysis (e.g., returning an appropriate reduction ratio given a desirable error bound).

I'm wondering how the proposed method scales to many real graphs. It would be helpful to know running times of the proposed method on different sizes of graphs.

It would be great if the authors can explain how the proposed method can be utilized in downstream tasks. Any specific examples/applications will be helpful to understand the practical value of the proposed method.

---

> ### Author Response · Authors · 2020-11-17
> **Responses to Reviewer2**
>
>
> First, thank you very much for your comments and feedback!
>
> #### Reduction ratio:
> This is a very good question. Right now we are following [2] on the choice of reduction ratio. Selecting the appropriate reduction ratio with controlled error would require a more rigorous theoretical analysis. Such theoretical analysis could be interesting future problems to study. Some challenges include:
>
> First, to talk about the performance of our model on test graphs, we will likely need to assume some generative models for graphs, say graphs sampled from graphons. The generalization error will depend both on the distributions of graphs, and the coarsening algorithm selected, which appears challenging.
>
> Second, even for the training graphs, what is the minimal loss we can achieve through the GNN is not very clear yet. If we don’t use a GNN learned approach, but use only an optimization-based approach (which cannot generalize to test graphs, that is, we have to run this expensive optimization procedure for each test graph), we have some theoretical analysis in appendix F.
>
> #### Scale and Runtime:
> In terms of the running time, note that once the neural network is trained, it needs very little additional time on test graphs (other than the node-coarsening algorithm we use).
>
> So the time is mostly on training our framework on training graphs. Here, most of the computation is spent on precomputing eigenvectors. But we only need to compute those eigenvectors once. Furthermore, as we showed in the paper, we could train our model on reasonably small graphs but apply to test graphs of much larger sizes. Note that we have already tested our GOREN framework to real graphs e.g, PubMed, Flickr etc.
>
> More concretely, the time complexity for training our GOREN framework in one batch is O(|E|k) where |E| is the number of edges and k is the number of eigenvectors. For synthetic graphs, it takes a few minutes to train the model. For real graphs like CS, Physics, PubMed, it takes around 1 hour. For the largest network Flickr of 89k nodes and 899k edges, it takes about 5 hours for most coarsening algorithms and reduction ratios. We also updated the time complexity part in appendix E.
>
> #### Downstream tasks:
> (1) In general, smaller representations of large graphs are easier for researchers to explore and to analyze their structures, e.g, to explore a huge graph by visualizing its coarsened graph. (2) The resulting graph can be a proxy for solving optimization problems on the original graph, e.g, multi-commodity flow. (3) Another downstream problem we had in mind is to apply our method to improve algebraic multigrid, partially motivated by [1].
>
> [1] Learning Algebraic Multigrid Using Graph Neural Networks https://arxiv.org/abs/2003.05744
>
> [2] Loukas, Andreas. "Graph Reduction with Spectral and Cut Guarantees." Journal of Machine Learning Research 20.116 (2019): 1-42.

---

### Decision · Program_Chairs · 2021-01-07
**Final Decision**

**Decision:**

Accept (Poster)

**Comment:**

This paper presents a way to use GNNs to learn edge weights of a coarsened graph given the node mapping from the original graph to the coarsened graph.  The paper is well-written and the approach is well-motivated as learning makes it easy to adapt the edge weights to different tasks and objectives, as illustrated in the graph Laplacian and Rayleigh quotient examples.  All the reviewers gage positive reviews for this paper, hence I recommend accepting this paper.

The reason for not promoting this paper further to spotlight or oral is that the paper addressed a relatively small problem, learning the edge weights given the node mapping, and the proposed method is quite simple.  Therefore this paper’s impact could be limited.

One suggestion to the authors is to present more results on downstream tasks, i.e. how does the proposed coarsening algorithm improve downstream task performance, instead of just losses defined without a downstream task in mind.  Example things to consider: does this approach improve graph classification accuracy?  Does this improve downstream GNN model’s efficiency without sacrificing accuracy?